# Mapping Post-Earthquake Landslide Susceptibility Using U-Net, VGG-16, VGG-19, and Metaheuristic Algorithms

**Mahyat Shafapourtehrany** [1], **Fatemeh Rezaie** [2,3,4], **Changhyun Jun** [5,*], **Essam Heggy** [6,7], **Sayed M. Bateni** [4], **Mahdi Panahi** [4], **Haluk Özener** [1], **Farzin Shabani** [8] **and Hamidreza Moeini** [9]

1 Kandilli Observatory and Earthquake Research Institute, Department of Geodesy, Bogazici University, Cengelkoy, Istanbul 34680, Turkey; mahyat.shafapour@boun.edu.tr (M.S.); ozener@boun.edu.tr (H.Ö.)
2 Geoscience Data Center, Korea Institute of Geoscience and Mineral Resources (KIGAM), 124, Gwahak-ro, Yuseong-gu, Daejeon 34132, Republic of Korea; rezaie@kigam.re.kr
3 Department of Geophysical Exploration, Korea University of Science and Technology, 217, Gajeong-ro, Yuseong-gu, Daejeon 34113, Republic of Korea
4 Division of Civil and Environmental Engineering and Water Resources Research Center, University of Hawaii at Manoa, Honolulu, HI 96822, USA; smbateni@hawaii.edu (S.M.B.); mpanahi@hawaii.edu (M.P.)
5 Department of Civil and Environmental Engineering, College of Engineering, Chung-Ang University, Seoul 06974, Republic of Korea
6 Viterbi School of Engineering, University of Southern California, Los Angeles, CA 90089, USA; heggy@usc.edu
7 Jet Propulsion Laboratory, California Institute of Technology, Pasadena, CA 91109, USA
8 Department of Biological and Environmental Sciences, College of Arts and Sciences, Qatar University, Doha P.O. Box 2713, Qatar; fshabani@qu.edu.qa
9 Faculty of Management and Economics, Science and Research Branch, Islamic Azad University, Tehran 1477893855, Iran; hr.moeini@srbiau.ac.ir
* Correspondence: cjun@cau.ac.kr

**Abstract:** Landslides are among the most frequent secondary disasters caused by earthquakes in areas prone to seismic activity. Given the necessity of assessing the current seismic conditions for ensuring the safety of life and infrastructure, there is a rising demand worldwide to recognize the extent of landslides and map their susceptibility. This study involved two stages: First, the regions prone to earthquake-induced landslides were detected, and the data were used to train deep learning (DL) models and generate landslide susceptibility maps. The application of DL models was expected to improve the outcomes in both stages. Landslide inventory was extracted from Sentinel-2 data by using U-Net, VGG-16, and VGG-19 algorithms. Because VGG-16 produced the most accurate inventory locations, the corresponding results were used in the landslide susceptibility detection stage. In the second stage, landslide susceptibility maps were generated. From the total measured landslide locations (63,360 cells), 70% of the locations were used for training the DL models (i.e., convolutional neural network [CNN], CNN-imperialist competitive algorithm, and CNN-gray wolf optimizer [GWO]), and the remaining 30% were used for validation. The earthquake-induced landslide conditioning factors included the elevation, slope, plan curvature, valley depth, topographic wetness index, land cover, rainfall, distance to rivers, and distance to roads. The reliability of the generated susceptibility maps was evaluated using the area under the receiver operating characteristic curve (AUROC) and root mean square error (RMSE). The CNN-GWO model (AUROC = 0.84 and RMSE = 0.284) outperformed the other methods and can thus be used in similar applications. The results demonstrated the efficiency of applying DL in the natural hazard domain. The CNN-GWO predicted that approximately 38% of the total area consisted of high and very high susceptibility regions, mainly concentrated in areas with steep slopes and high levels of rainfall and soil wetness. These outcomes contribute to an enhanced understanding of DL application in the natural hazard domain. Moreover, using the knowledge of areas highly susceptible to landslides, officials can actively adopt steps to reduce the potential impact of landslides and ensure the sustainable management of natural resources.

**Keywords:** earthquake; landslide detection; deep learning; landslide susceptibility mapping; Japan

## 1. Introduction

Landslides are potentially devastating natural hazards that can occur as a secondary disaster during or after an earthquake event [1]. Traditionally, earthquake-induced landslides have been recognized and mapped through field surveys [2]. However, such methods involve several limitations, such as difficulties in accessing remote regions, high risks associated with entering the affected areas, and high cost and time requirements, and thus, scientists have actively sought to identify innovative methods to address these problems [3]. With advancements in geospatial technology, different approaches have been developed to detect earthquake-induced landslides [4,5]. For example, landslide patterns can be recognized by remote sensing (RS) technologies [6]. RS images capture a large area without any access limitations or safety concerns. The objective of this study was to detect the 2018 Iburi earthquake-induced landslide areas and map their susceptibility. To this end, it was necessary to select the proper data and methodology.

As mentioned, RS and geospatial technologies can provide a vast range of data for large areas, with high resolution and short visit times [7]. Satellite imagery from one [8] or multiple sensors [9] can be used in landslide detection, depending on the aim and scope of the research. Cost and data availability are other factors affecting the data selection. The existing studies on landslide detection have used RapidEye satellite imagery [10], ALOS-PALSAR [11], hyperspectral imagery [12], synthetic aperture radar images [13], Landsat images [14], Sentinel images [15], and unstaffed aerial vehicle images [16].

In this context, one of the simplest RS approaches is to detect the landslides manually through the visual interpretation of aerial photographs [17]. However, this approach is time-consuming and requires expert knowledge and intensive resources [18]. Recent approaches mostly consist of two stages: (1) the landslides' representative features (either pixel or object-based) are extracted, and (2) the features are classified as landslide or non-landslide pixels or segments [19]. Several techniques can be found in the literature, including but not limited to principal component analysis (PCA) [20], maximum likelihood classification (MLC) [21], wavelet transform [22], and time-series analysis [23]. Each of these methods has its own advantages and disadvantages. Among the existing methods, machine learning (ML) techniques have been widely applied in landslide detection [19,24], with support vector machine (SVM) and random forest (RF) [25–27] methods being commonly used.

Notably, the application of deep learning (DL) as a subdivision of ML remains inadequately explored. Ghorbanzadeh et al. [28] reviewed several DL methods used in landslide detection and recognition and noted that since 2019, DL algorithms, such as convolutional neural networks (CNNs), have been increasingly applied in landslide inventory mapping. The widespread use of DL is likely related to its advantages, such as the lack of requirement of expert opinion, intensive supervision, and fieldwork [29]. In addition, to use conventional methods such as SVM and RF, interpretation logic needs to be designed, which may increase the algorithm complexity [20]. In contrast, in DL techniques, data are processed in several stages, the amount of data is reduced through mapping relationships, and significant characteristics of data are extracted [30]. Unlike conventional ML methods that cannot effectively process natural data in their raw form, DL techniques can use raw data as an input and learn highly complex functions [31].

Ye et al. [12] performed one of the pioneering studies involving the use of DL for detecting landslide areas based on hyperspectral RS. Specifically, the authors used the DL framework with constraints (DLWC) to recognize and detect landslide areas. The results highlighted that DLWC outperformed other conventional methods. Representative examples of DL methods include the dense convolutional network (DenseNet) [32], U-Net, residual U-Net (ResUNet), generative adversarial networks, and deep convolutional neural networks (DCNNs), which have been extensively evaluated by researchers. For instance, Ghorbanzadeh et al. [33] applied U-Net and ResUNet to Landsat and Sentinel-2 imageries to compare their landslide detection and recognition performances in three regions of Western Taitung County. Yi and Zhang [8] developed LandsNet to map the landslide inventory from two areas in Sichuan province, China. The performance of the proposed method

was evaluated in comparison with ResUNet and DeepUNet. LandsNet, with an accuracy of 86.89%, outperformed the other two algorithms. Su et al. [34] developed a deep CNN model named LanDCNN to map the landslide extent in Lantau Island, Hong Kong. To this end, a 0.5 m digital terrain model and high-resolution bitemporal RGB aerial photos were used. Among DL methods, U-Net is considered to be a robust algorithm. Meena et al. [35] applied U-Net, RF, SVM, and the K-nearest neighbor models to two datasets from different sources (RapidEye and ALOS-PALSAR) to compare their potential in landslide detection. The first dataset included optical bands of RapidEye imagery, and the second dataset involved topographical data derived from ALOS-PALSAR. The researchers highlighted that U-Net generated the most accurate outcomes.

In addition to generating the earthquake-induced landslide inventory map, it is necessary to detect landslide-prone areas to implement appropriate measures for preventing or minimizing the potential impacts of landslides [36]. Several researchers have applied statistical methods to map landslide-susceptible regions [37–39]. Merghadi et al. [40] explored the top-cited research and review articles pertaining to the landslide susceptibility domain. Similarly, Reichenbach et al. [41] and Shano et al. [42] reviewed the most popular and well-known approaches in this domain. Most of the existing studies were noted to use RF [43], logistic regression [44], artificial neural network [45], SVM [37], decision tree [46], K-nearest neighbors [47], weights of evidence [48], adaptive neuro-fuzzy inference systems [49], and evidential belief functions [50]. In general, landslides have complex structures due to numerous inherent causative and external triggering factors [51]. Therefore, it is necessary to identify more robust techniques for mapping landslide-prone regions.

ML techniques have been noted to outperform traditional statistical methods [52,53], and hybrid and ensemble techniques have been proven to outperform standalone ML models [54]. In recent times, the research domain of landslide susceptibility has witnessed the incorporation of diverse DL methodologies. For instance, the integration of meta-heuristic optimization algorithms such as gray wolf optimizer (GWO) with ML can help enhance the predictive performance of the model, as highlighted by Hakim et al. [55], who integrated CNN with GWO and imperialist competitive algorithm (ICA) to map the landslide-susceptible zones in Icheon City, South Korea. The authors concluded that ensemble methods of DL and optimization algorithms can produce more reasonable outcomes for landslide susceptibility mapping.

To the best of our knowledge, U-Net, VGG-16, and VGG-19 algorithms have not been widely utilized for earthquake-induced landslide detection, in comparison with more conventional methods such as PCA [56–58] and MLC [59]. Therefore, the objective of this study was to evaluate the performances of these algorithms in detecting the locations affected by the landslide induced by the 2018 Iburi earthquake, based on Sentinel-2 satellite imagery. U-Net, VGG-16, and VGG-19 were selected, owing to their superiority to conventional ML methods, as these methods can capture both local and global features in the image, overcome the problem of vanishing gradients, capture more complex features because of their deeper architecture, and be trained on a small dataset. Moreover, U-Net, VGG-16, and VGG-19 have a simple structure, high training speed, and high classification accuracy [60–63]. The inventory dataset generated from the first stage was used in landslide susceptibility mapping. Moreover, the standalone CNN model and that hybridized by the ICA and GWO algorithms were used to create the landslide susceptibility maps. Specifically, the CNN was integrated with ICA and GWO to tune its hyperparameters. ICA and GWO have been noted to be better than other metaheuristic algorithms in terms of the convergence rate, capability of simultaneously optimizing multiple hyperparameters, robustness, scalability, and ability to balance exploration and exploitation during the search process, thereby preventing the model from being trapped in local optima [64,65]. The predictive ability of DL algorithms was assessed using statistical indices. Moreover, the correlation between landslide occurrences and geo-environmental factors was scrutinized.

The contributions of this study to the literature are four-fold: (1) The efficiency of three DL algorithms (namely, U-Net, VGG-16, and VGG-19) in detecting earthquake-induced

landslides, based on Sentinel-2 satellite imagery, is assessed; (2) spatial predictions of landslide susceptibility are generated using the CNN algorithm; (3) the feasibility of the ICA and GWO metaheuristic algorithms in optimizing the hyperparameters of the standard CNN for enhancing its predictive performance and reliability of the results is evaluated; and (4) the outcomes can enhance our knowledge of landslide modeling by facilitating the identification of the topographic, hydrological, and anthropogenic factors that most notably influence the mapping of landslide-susceptible areas. The findings are expected to facilitate further scientific research and the formulation of natural hazard management plans.

## 2. Methodology

The process flow of this research is illustrated in Figure 1. There are two main stages: landslide detection and landslide susceptibility mapping. Landside detection is performed using the U-Net, VGG-16, and VGG-19 algorithms, as discussed in Section 2.2. The F1-score, precision, and recall values are used to assess the landslide inventories. The highest-performing algorithm is used to detect landslides over the study area. A total of 63,360 cells are detected as landslide locations. An equal number of locations are randomly detected as non-landslide data within the areas characterized by minimal slopes within lower elevation areas, in which a landslide has been considered unlikely to occur [55]. The final inventory locations (non-landslides and landslides) are divided into training and testing zones in a ratio of 70:30 for landslide susceptibility mapping. The testing and training subsets are overlaid with rasterized influencing factors to extract the attributes of non-landslide and landslide samples. In the landslide susceptibility mapping stage, the frequency ratio (FR) is used to evaluate the spatial correlation among landslide locations and landslide-susceptibility-related factors. Afterward, the three DL models are developed in MATLAB software (Version: R2022b, https://www.mathworks.com/) (Accessed on 11 September 2023). The FR generated for each class of every landslide-susceptibility-related factor is input to the models to create landslide susceptibility maps using DL algorithms; i.e., the CNN, CNN-ICA, and CNN-GWO methods. The output of the models is a landslide susceptibility map that shows landslide-susceptible regions in the entire study area. To optimize the performance of the CNN model, the hyperparameters are fine-tuned by minimizing the root mean square error (RMSE) between the predicted and actual values. The performances are evaluated using the area under the receiver operating characteristic curve (AUROC) and RMSE.

### 2.1. Study Area

The Iburi earthquake occurred on 6 September 2018, in the Iburi region of Hokkaido, Northern Japan [66] (Figure 2). The epicenter was located at 42.690°N by 142.006°E, 300 km from the southeastern Kuril Trench [67]. According to the Japan Meteorological Agency (JMA), the Iburi earthquake had a maximum seismic intensity of 7.0, a moment magnitude (Mw) of 6.6, and a JMA magnitude of 6.7 [68]. Until 31 October 2018, 311 aftershocks (maximum seismic intensity higher than 1.0) were recorded by JMA [69]. This earthquake caused severe social and economic damage to Hokkaido and all of Japan [70]. According to the Ministry of Internal Affairs and Communications, Japan, 394 houses collapsed and 1061 were damaged. In addition, 41 people were killed, and 691 people were injured. Approximately 82% of the total fatalities resulted from shallow landslides in Atsuma. Most of these landslides were shallow earth slides that occurred approximately 20 km north of the epicenter, likely because of the thickness of the surface geology (between 4 and 5 m) in the region [71]. The remaining landslides were deep rockslides that occurred within 10 km of the epicenter. Pyroclastic flow deposits and pyroclastic fall deposits covered 40% of Hokkaido Island. The main reason for earthquake- and rainfall-induced landslides in Hokkaido was the presence of Quaternary volcanic soils in the region [72]. The powerful typhoon Jebi, which hit the study area a day before the earthquake [73], adversely affected the slope stability of this region as well. Rainfall of 10–20 mm was recorded by

the Automated Meteorological Data Acquisition System (Meteorological Agency of Japan, https://www.jma.go.jp/) (Accessed on 11 September 2023).

### 2.2. Data

#### 2.2.1. Landslide-Detection-Related Dataset

Level-1C Sentinel-2 imagery obtained on 10 September 2018, was used to generate landslide inventory maps via U-Net, VGG-16, and VGG-19 for image segmentation and landslide detection. The algorithms are described in Section 2.3. Atmospheric correction was performed using the SNAP software (Version: 9.0.0, https://www.step.esa.int/) (Accessed on 11 September 2023) eveloped by the European Space Agency for data processing [74]. First, the study area was divided into three zones: training, testing, and prediction zones (Figure 3). The training and testing zone data were used to train and evaluate the algorithms, respectively. The most accurate algorithm among the three algorithms was then used to detect the landslide locations in the prediction zone. The process was initiated by digitizing a number of landslide samples from the training zone to be used in training U-Net, VGG-16, and VGG-19. To prepare the inventory layer (independent factor) for landslide susceptibility processing, the same number of non-landslide locations were selected. Values of 1 and 0 were assigned to landslide and non-landslide locations, respectively. All locations were split into training (70%) and testing (30%) groups [75,76].

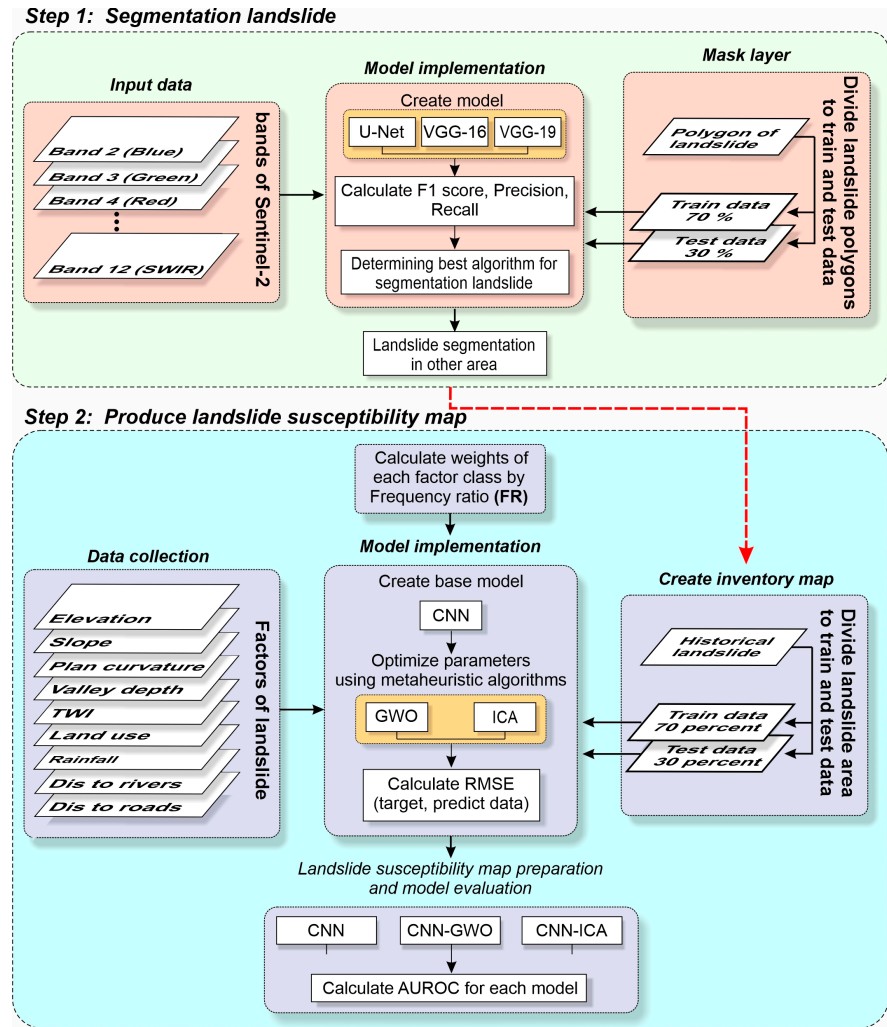

**Figure 1.** Process flow of the research, representing the model development for landslide detection and susceptibility mapping.

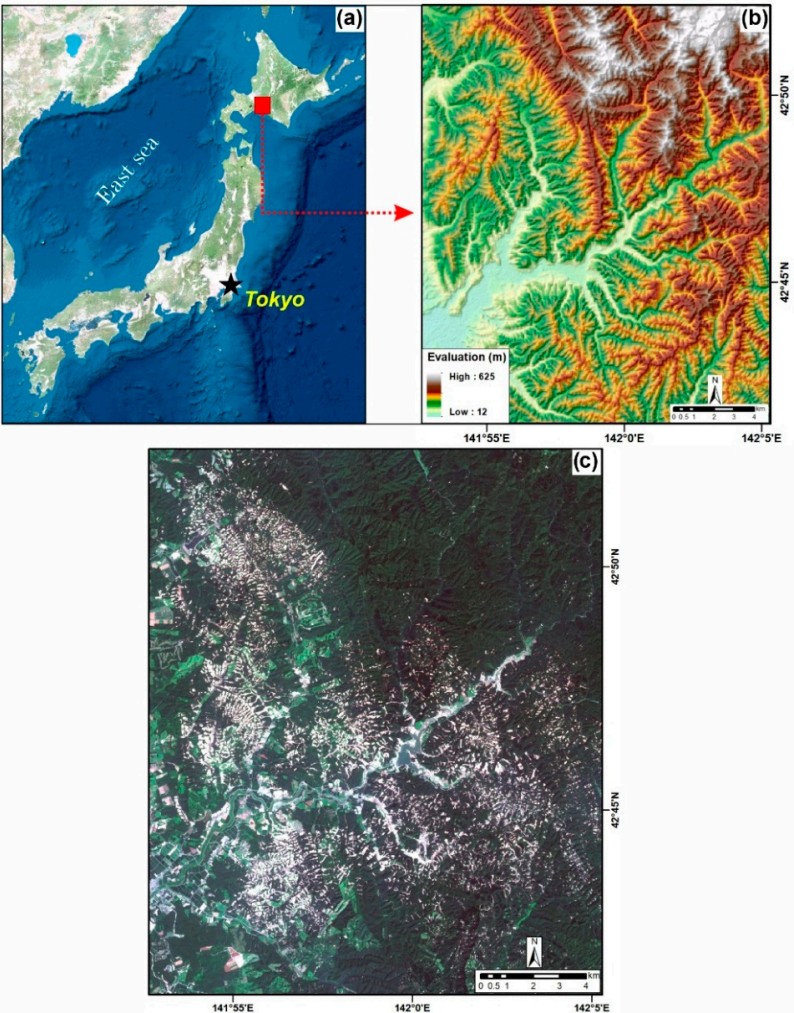

**Figure 2.** Study area: (**a**) location of Japan capital (black star symbol) and Iburi region of Hokkaido, Northern Japan (red square symbol), (**b**) The digital elevation map, and (**c**) location of landslides occurred during and after the 2018 Iburi earthquake (white symbol).

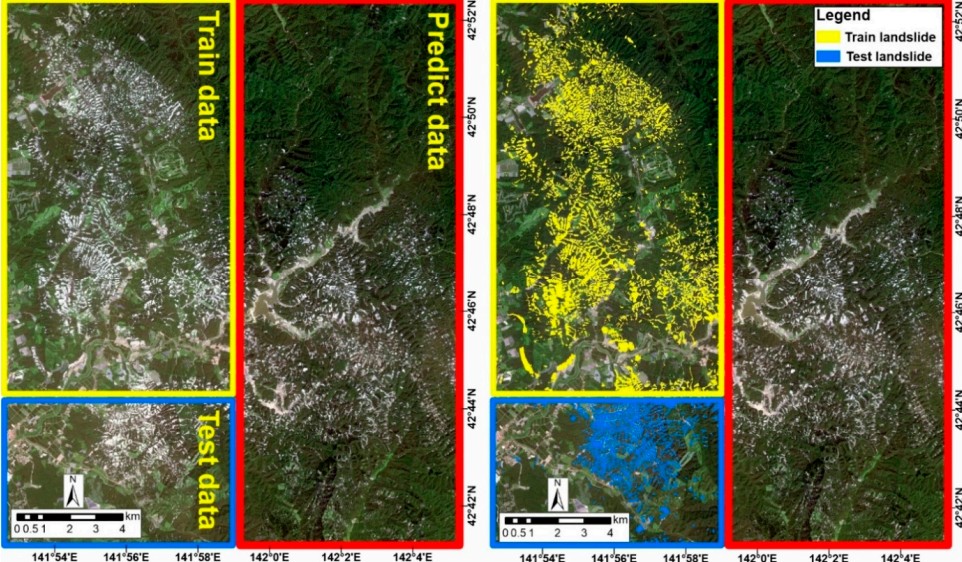

**Figure 3.** Training, testing, and predicted zones on a Sentinel-2 image.

### 2.2.2. Factors Influencing Landslide Susceptibility

To date, there is no predefined framework for the selection of factors influencing earthquake-induced landslides [77]. Researchers typically select factors with reference to the literature and expert knowledge. In this study, nine factors, i.e., elevation, slope, plan curvature, valley depth, topographic wetness index (TWI), land cover, rainfall, distance to rivers, and distance to roads (Figure 4), were selected based on a literature review to be used in landslide susceptibility mapping [78–80].

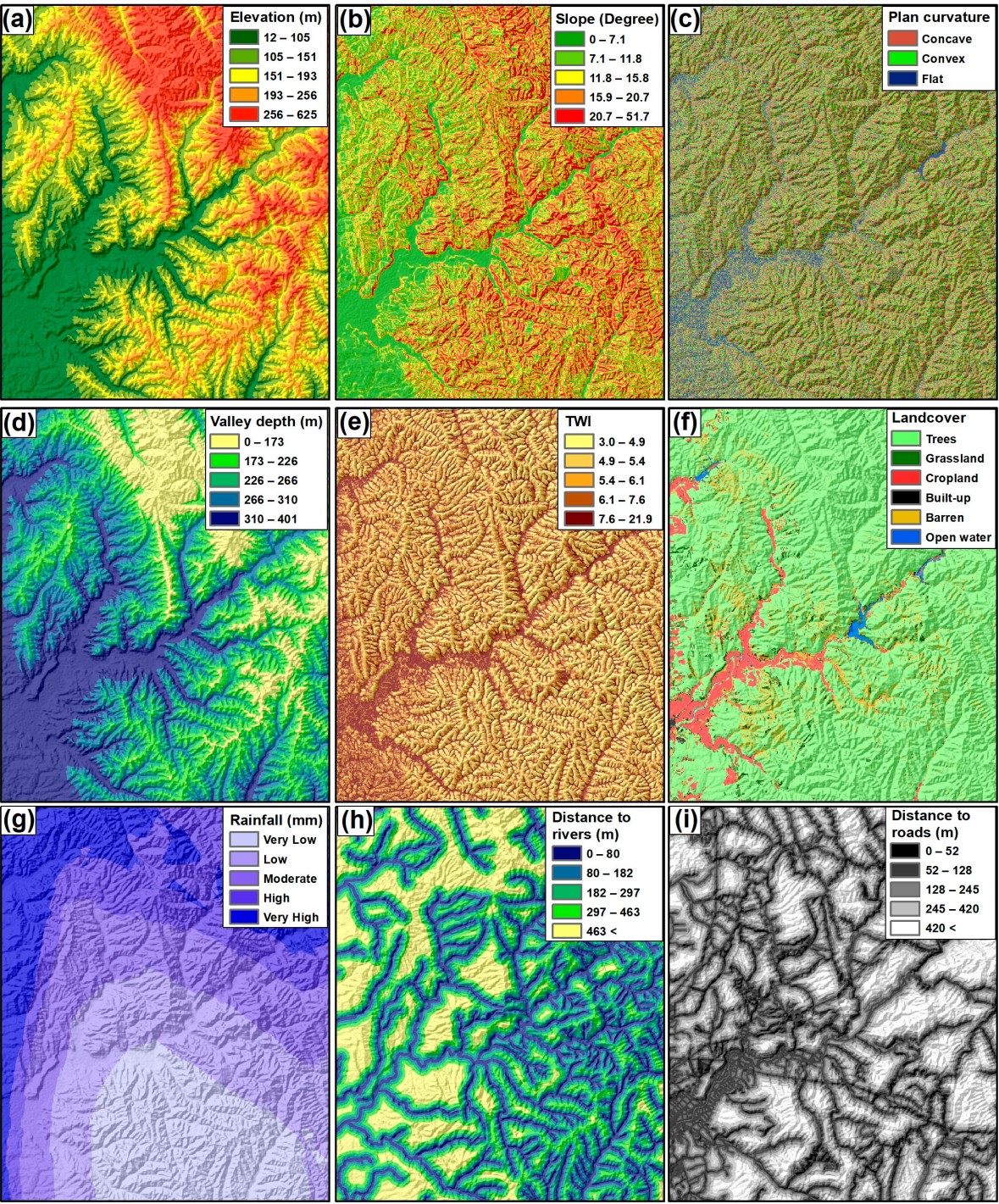

**Figure 4.** Spatial database of landslide susceptibility-related factors: (**a**) elevation, (**b**) slope, (**c**) plan curvature, (**d**) valley depth, (**e**) TWI, (**f**) land cover, (**g**) rainfall, (**h**) distance to rivers, and (**i**) distance to roads.

**Elevation**. Surface topography considerably affects the density and spatial extent of landslides by controlling the flow direction and soil moisture [81]. The minimum and maximum heights of the region can be defined using the elevation. A digital elevation model (DEM) with a spatial resolution of 30 m was downloaded from the United States Geological Survey website (Figure 4a). Other factors, i.e., the slope (Figure 4b), plan curvature (Figure 4c), valley depth (Figure 4d), and TWI (Figure 4e) were derived from the DEM.

**Slope**. The slope, which defines the steepness of the ground, affects landslide occurrence [82] owing to its influence on the moisture concentration, land instability, and hydraulic continuity.

**Plan Curvature**. The curvature indicates the land geometry and slope variation [83]. Negative, positive, and zero curvatures indicate concave, convex, and flat regions, respectively. Ohlmacher [84] highlighted the strong correlation between landslide susceptibility, type of landslide, and plan curvature of the region.

**Valley Depth**. The valley depth refers to the vertical distance of a point from the base level of the channel network [85]. This factor controls the slope stability, transportation and accumulation of water, as well as the weathering process, thereby affecting landslide occurrences [86].

**TWI**. The TWI indicates the soil moisture [87]. Landslide occurrence probability is negatively related to the TWI. Specifically, landslide susceptibility increases as cohesion decreases due to moisture loss.

**Land cover**. Land cover represents human activities and land cover variations, which considerably affect landslide occurrences [88]. For instance, the presence of vegetation may increase water accumulation, thereby decreasing the slope stability [89]. In this study area, the land cover cases involve trees, grassland, cropland, built-up area, barren land, and open water (Figure 4f). The land cover map for 2018 was obtained from the Sentinel-2A satellite image operated by the European Space Agency.

**Rainfall**. Rainfall intensity considerably affects landslide occurrences. Li et al. [90] comprehensively analyzed the effect of rainfall on earthquake-induced landslide occurrence. The mean annual rainfall map (2015–2020) was generated by the Climate Hazards Group infraRed Precipitation with Stations dataset (https://chc.ucsb.edu/data/chirps) (Accessed on 11 September 2023) (Figure 4g).

**Distance to Rivers**. Rivers may generate cuts in rocks. Moreover, the distance from rivers affects the slope stability as the saturation degree of the slope-forming materials varies with this distance [86]. According to Huang et al. [91], the probability of landslide occurrence increases with the proximity to rivers. Figure 4h shows the map of the distance to rivers.

**Distance to roads**. Roads are often the sources of slope instability and continuity. The flow of water can be altered by a road segment, which can function as a barrier, source, sink, or corridor, and its location in the mountains often results in it triggering landslides. The Open Street Map portal (https://www.openstreetmap.org/) (Accessed on 11 September 2023) was used to identify the road and river locations (Figure 4i). The distances to rivers and roads were measured in the ArcGIS Environment using the "Euclidean distance" spatial analyst tool.

## 2.3. Landslide-Detection

### 2.3.1. U-Net

CNN is a form of DL that has the unique ability to take raw image data as input, eliminating the need for the conventional and intricate processes of image preprocessing and complex feature extraction [92]. U-Net is a CNN architecture that was designed in 2015 for pixel-based classification [93]. Unlike regular CNNs, which classify an image as a whole, U-Net classifies each pixel [94]. This process is known as semantic segmentation and is commonly used in biomedical and military applications [93]. An image is input to U-Net, and a segmented image is output. U-Net has two parts: an encoder that reduces the image

size and a decoder that increases it, resulting in a segmented image with the same size as the input image [35]. The U-Net model consists of two parts, each with five levels. The first part, the encoder, reduces the dimensions of the input image by half at each level through two convolution layers, a max-pooling layer, and a dropout layer to avoid overfitting. The second part, the decoder, reverses this process and increases the image dimensions using information from corresponding levels in the encoder. This process results in an output image with the same size as the input and same number of classes [93]. The scalability of the architecture and use of data augmentation render it effective even with small training sets. Figure 5 shows the network structure of U-Net.

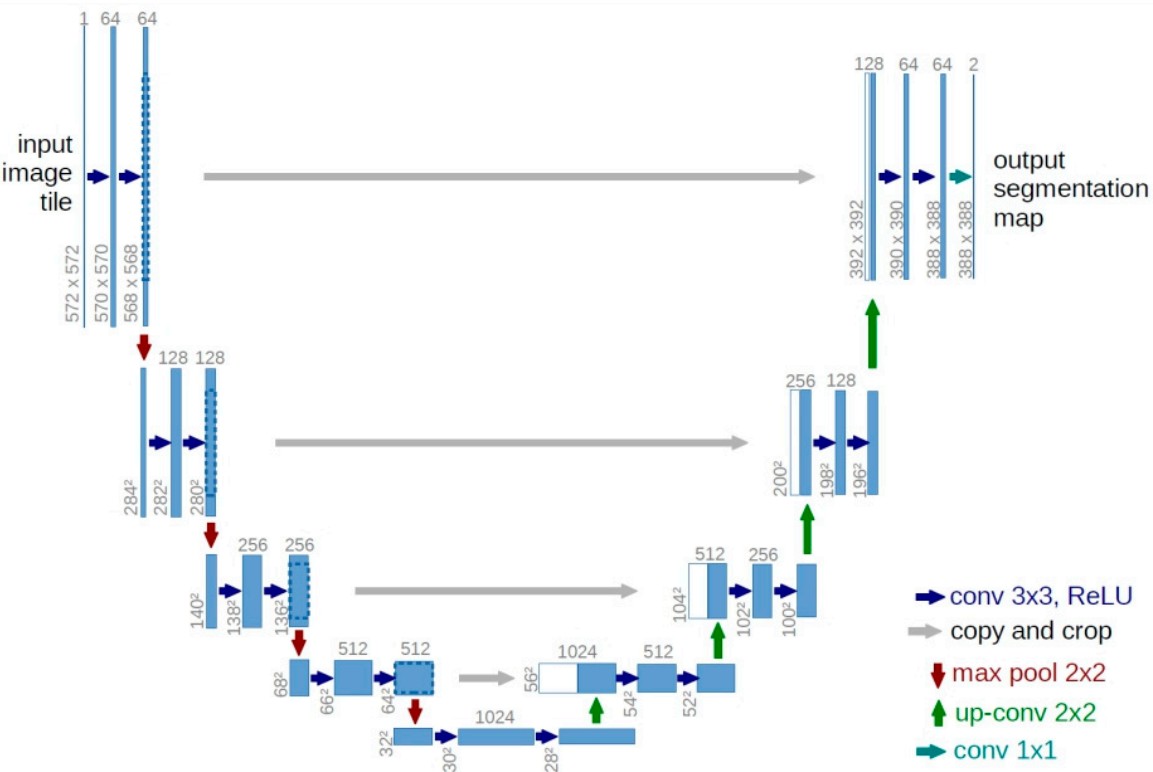

**Figure 5.** Network structures of U-Net.

### 2.3.2. VGG Models

VGG models outperform the prior models in terms of both speed and accuracy [93], owing to their deeper architecture and the use of pre-trained models. Furthermore, the incorporation of additional layers with smaller kernels increases the nonlinearity of the model, which is a key factor for successful DL [95]. VGG-16 is a 16-layer CNN that has been pre-trained using over a million images from the ImageNet database [96]. The model can recognize 1000 distinct object categories and consists of a combination of convolution and max-pooling layers [97]. The network culminates in two fully connected layers (FC), followed by SoftMax activation for prediction [98]. With approximately 138 million parameters, VGG-16 is a comprehensive model [99]. VGG-19 is a 19-layer DCNN that has been pre-trained on over a million images from the ImageNet database [100]. Similar to VGG-16, this model can also classify images into 1000 object categories, including objects such as keyboards, pencils, mice, and other animals. Consequently, VGG-19 can learn rich feature representations for a diverse range of images. The sizes of the input images for VGG-16 and VGG-19 are 224 × 224. Figure 6 shows the network structures of VGG-16 and VGG-19.

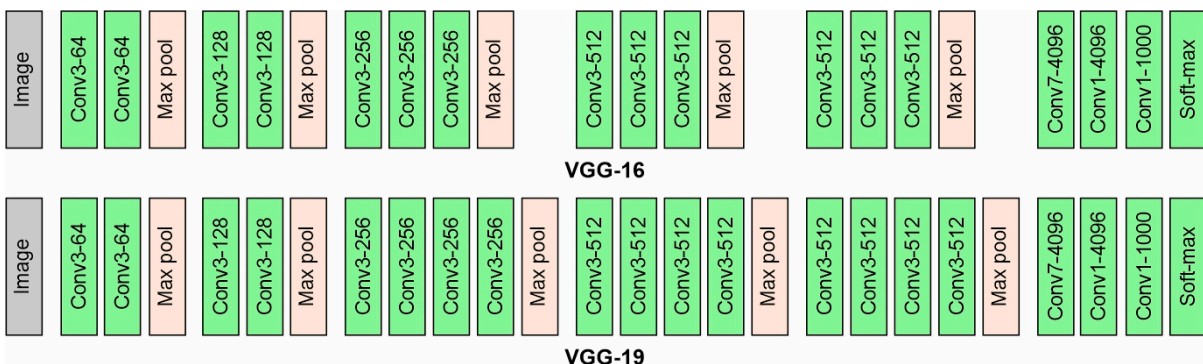

**Figure 6.** Network structures of VGG-16 and VGG-19.

*2.4. Landslide Susceptibility Mapping*

2.4.1. FR

The FR is used to evaluate the correlation between each class of every landslide susceptibility-related factor and landslide occurrences [101]. Equation (1) is used to calculate the proportion of pixels for each landslide factor to the pixels corresponding to landslides.

$$FR = \frac{\% \text{ pixel of landslide points}}{\% \text{ pixel class of landslide effective factor}} \tag{1}$$

2.4.2. CNN

CNN is a DL algorithm that uses neural networks with layers for convolution, activation, and pooling, along with an FC layer [102]. The convolutional layers identify connections between feature classes using input image features for enhancing the performance [103]. Nonlinear results from the convolutional process are processed in the activation layer with the activation function being the rectified linear unit to enhance the nonlinear properties of the neural network [104]. The pooling of layers after activation helps prevent overfitting by reducing the dimension of feature classes through downsampling [105]. The final layer, the FC layer, combines the network from pooling into one neuron to produce the final output [106]. The CNN generates a nonlinear mapping from the input to the output without the need for defining precise mathematical expressions between the input and output [107]. A detailed description of the CNN algorithm can be found in [108,109].

2.4.3. ICA

The ICA is a metaheuristic optimization algorithm that simulates imperialistic competition to model national socioeconomic trends [110]. The starting population is referred to as countries or empires, consisting of imperialists and colonies. The countries with low costs are selected as imperialist countries, and colonies are divided among them based on the power of the imperialists [65]. Competition starts when the empires grow and the imperialists control a group of colonies, gaining more through their dominance [111]. The positions of imperialists and colonies can be interchanged if the colonies become stronger, resulting in the elimination of the losing empire and rise of the colonies as the dominant country [112]. A detailed description of the ICA algorithm can be found in [65].

2.4.4. GWO

The GWO is a metaheuristic optimization algorithm that models the social hierarchy and hunting behavior of gray wolves [113]. The gray wolf pack is divided into four levels: alpha ($\alpha$), beta ($\beta$), delta ($\delta$), and omega ($\omega$) [114]. Prey hunting consists of encircling, hunting, and attacking [115]. The $\alpha$ wolf leads the pack with assistance from the $\beta$ wolves. The $\delta$ wolves are responsible for tasks such as scouting, guarding, and hunting, whereas the $\omega$ wolves occupy the lowest rank and serve as scapegoats [116]. To mathematically

model the social hierarchy of wolves, the fittest solution is designated as α, and the second and third best solutions are β and δ, respectively. The remaining candidate solutions are grouped as ω. The hunting (optimization) is guided by α, β, and δ wolves, and the ω wolves follow their lead. The encircling behavior of gray wolves during a hunt is defined as

$$\vec{D} = \left| \vec{C} \cdot \vec{X}_p(t) - \vec{X}(t) \right| \qquad\qquad \vec{C} = 2 \cdot \vec{r}_2 \qquad\qquad (2)$$

$$\vec{X}(t+1) = \vec{X}_p(t) - \vec{A} \cdot \vec{D} \qquad\qquad \vec{A} = 2\vec{a} \cdot \vec{r}_1 - \vec{a} \qquad\qquad (3)$$

where $\vec{X}_p$ and $\vec{X}$ are the position vectors of a gray wolf and the prey at the current iteration (t), respectively. $\vec{C}$ and $\vec{A}$ are coefficient vectors, and $\vec{r}_1$ and $\vec{r}_2$ are random vectors in [0, 1]. $\vec{C}$ is the component that favors exploration and local optima avoidance. Due to the lack of knowledge regarding the location of the prey (optimum solution) in an abstract search space, it is assumed that α, β, and δ know the potential location of prey, and the other wolves update their positions according to the positions of the α, β, and δ wolves, as follows:

$$\vec{D}_\alpha = \left| \vec{C}_1 \cdot \vec{X}_\alpha - \vec{X} \right|, \ \vec{D}_\beta = \left| \vec{C}_2 \cdot \vec{X}_\beta - \vec{X} \right|, \ \vec{D}_\delta = \left| \vec{C}_3 \cdot \vec{X}_\delta - \vec{X} \right| \qquad (4)$$

$$\vec{X}_1 = \vec{X}_\alpha - \vec{A}_1 \cdot \left( \vec{D}_\alpha \right), \ \vec{X}_2 = \vec{X}_\beta - \vec{A}_2 \cdot \left( \vec{D}_\beta \right), \ \vec{X}_3 = \vec{X}_\delta - \vec{A}_3 \cdot \left( \vec{D}_\delta \right) \qquad (5)$$

$$\vec{X}(t+1) = \frac{\vec{X}_1 + \vec{X}_2 + \vec{X}_3}{3} \qquad\qquad (6)$$

Finally, the wolves attack the prey when $\left| \vec{A} \right|$ is lower than 1. When $\left| \vec{A} \right| > 1$, the wolves diverge from one another to search for prey [114].

### 2.5. Accuracy Assessment
2.5.1. Performance Assessment of Landslide Detection Algorithms

The landslide detection accuracy is evaluated using the precision, recall, and F1-score [117]. The precision indicates the percentage of the correctly detected landslide pixels to the total number of samples predicted as landslides. Recall refers to the proportion of landslides that the algorithm correctly identifies out of the total pixels actually representing landslides. A high precision indicates that the model predicts fewer non-landslide areas as landslides, and a high recall indicates that the model misses fewer landslide areas [118]. A model with high precision but low recall may miss several landslide areas, whereas a model with high recall but low precision may predict many non-landslide areas as landslides. Therefore, a balance between precision and recall is desirable [34,118,119]. The F1-score represents the harmonic average among precision and recall, as these parameters assess different aspects of the model. A higher F1-score (maximum value = 1) corresponds to superior landslide detection capability. The performance metrics are defined as follows [120]:

$$\text{Precision} = \frac{TP}{TP + FP} \qquad\qquad (7)$$

$$\text{Recall} = \frac{TP}{TP + FN} \qquad\qquad (8)$$

$$\text{F1} - \text{score} = 2 \times \frac{\text{Precision} \times \text{Recall}}{\text{Precision} + \text{Recall}} \qquad\qquad (9)$$

where TP and FP represent the numbers of pixels correctly and incorrectly classified as a landslide, respectively. FN indicates the number of pixels mistakenly identified as non-landslide.

### 2.5.2. Assessment of Landslide Susceptibility Maps

The accuracy of the landslide susceptibility maps is assessed using the RMSE and AUROC. AUROC is the most widely used validation parameter in natural hazard susceptibility assessment [121]. This term represents the area enclosed by the receiver operating curve, with values ranging from 0 to 1. A larger area corresponds to a more accurate model, as it depicts the relationship between specificity and sensitivity [122]. The fitting and predictive performance of the models is evaluated based on the training and testing data, respectively.

## 3. Results

### 3.1. Correlation among the Landslide Location and Related Factors

The FR is used to define the correlation between the landslide occurrences and each class of every landslide influencing factor. Figure 7 shows the measured values. Values greater than 1 indicate a stronger impact on landslide occurrence.

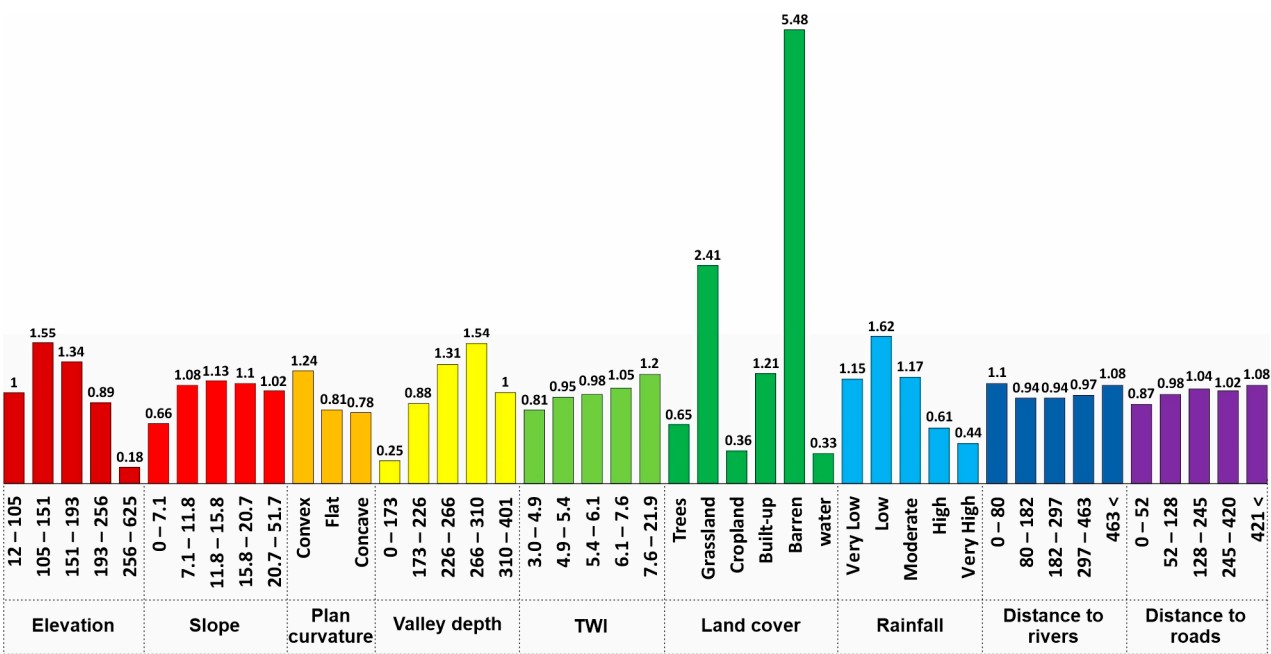

**Figure 7.** Frequency ratios of landslide-related factors.

In terms of the elevation, areas with an elevation of 105–193 m receive the highest FR, likely because of gravity and other factors [123]. In terms of the slope factor, steeper slopes are associated with a higher landslide occurrence probability [124], likely because of increased shear stress and the effect of gravity. In addition, sharp slopes are associated with lower soil thicknesses [125]. In terms of the curvature, only the convex class receives an FR greater than 1. FR values for the valley depth indicate that landslides are more likely to occur at larger valley depths (>226). In terms of the TWI, the FR values linearly increase from the lowest to the highest class. This finding is supported by other studies, such as that of Yilmaz [126], who reported that landslides typically occur at higher TWIs. In terms of the land cover, classes of grassland, built-up areas, and barren areas receive the highest FR values of 2.41, 1.21, and 5.48, respectively. The lowest classes of rainfall (very low, low, and moderate) receive higher FR values, showing the inverse correlation between landslide occurrence and rainfall. In the case of distance to rivers, the highest FR value of 1.1 corresponds to the smallest distance of 0–80 m. Unlike the distance to rivers, the highest FR value (1.08) for the distance to roads corresponds to the largest distance of >421 m.

*3.2. Landslide Detection*

U-Net, VGG-16, and VGG-19 are trained using the training dataset from the training zone (Figure 3). After being trained, the algorithms are applied to the test zone (Figure 3) to detect landslides and predict their possible locations. Figure 8 shows the predicted landslides in the test zone, along with the real landslides for visual comparison.

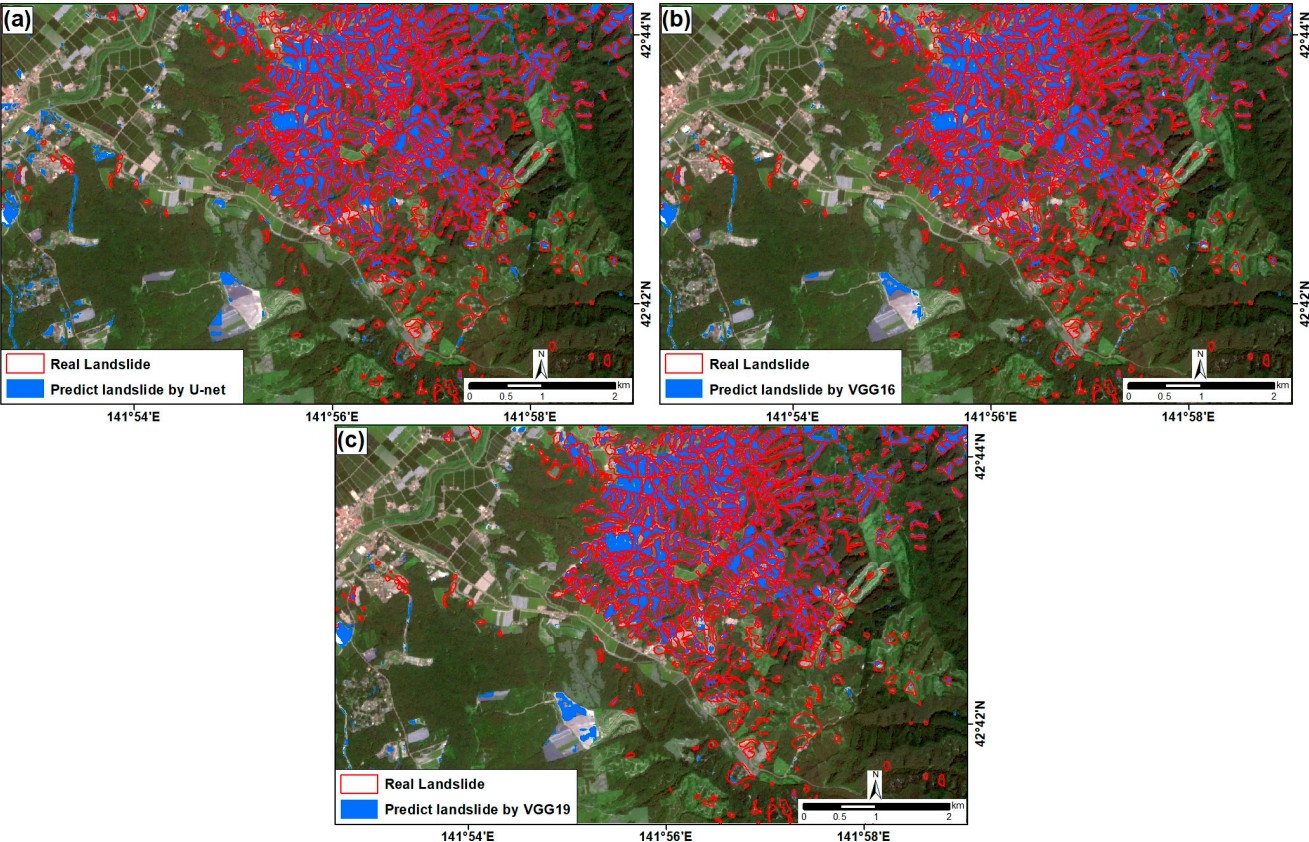

**Figure 8.** Real and predicted landslides in the test area by (**a**) U-Net, (**b**) VGG-16, and (**c**) VGG-19.

After training the algorithms and testing their performances, they were applied to the prediction zone (Figure 3). Figure 9 shows the outputs of the different algorithms.

The performances of the algorithm were evaluated using the precision, recall, and F1-score. Figure 10a shows the accuracy assessment metrics for each algorithm in the training stage. VGG-16 and VGG-19 receive the highest precision value of 0.85. The best recall (0.81) corresponds to U-Net. The F1-score is comparable for all three algorithms. Although U-Net achieves the lowest precision in the training step, it achieves the highest precision (0.85) in the test step (Figure 10b). U-Net also achieves the lowest recall and F1-score.

Figure 11 illustrates the validation loss curves for U-Net, VGG-16, and VGG-19. The loss curves of all algorithms are similar, indicating improvements during the later steps of the training. The accuracy of all three algorithms increases as the amount of loss decreases.

*3.3. Landslide Susceptibility Mapping*

VGG-16 yields the most accurate landslide inventory. Therefore, it is used to perform earthquake-induced landslide susceptibility mapping by integrating CNN, CNN-ICA, and CNN-GWO. The outputs derived from each method are classified into very low, low, moderate, high, and very high susceptibility zones (Figure 12). To this end, the quantile classification scheme is used, which has been commonly applied by researchers in the natural hazard domain. The least overestimation can be observed in the very high class in the landslide susceptibility map derived by the CNN-ICA. Figure 13, which

shows the percentage of area in each susceptibility class, confirms this observation. The landslide susceptibility of various geographic regions exhibits variability. The northern areas demonstrate a low susceptibility to landslides, whereas the central regions situated near roads and rivers are characterized by a higher predisposition to landslides (Figure 12). As shown in Figure 13, the high susceptibility class in the maps derived from CNN and CNN-ICA cover the largest area. According to the CNN model, most of the study area has elevated levels of susceptibility to landslides, with 28.68% and 20.22% of the regions classified as those with high and very high susceptibility, respectively. The CNN-GWO model indicates that lower proportions (approximately 20.34% and 18.08%, respectively) of areas exhibit high and very high susceptibility to landslide incidents. The regions with very high susceptibility must be prioritized for deploying landslide risk mitigation strategies, such as improved monitoring, introduction of early warning systems, and implementation of land-use regulations aimed at restricting development in these vulnerable regions.

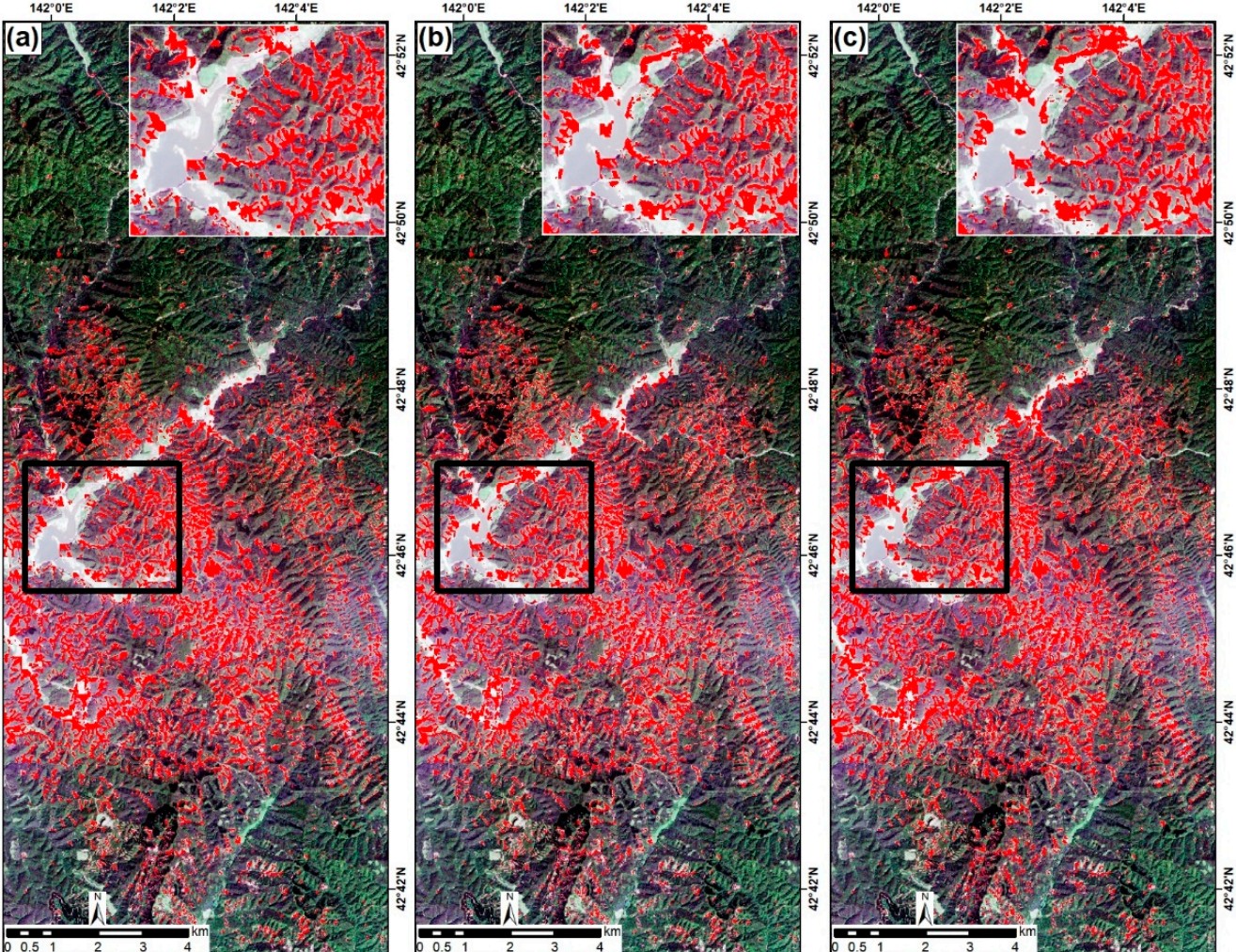

**Figure 9.** Final segmentation results derived from (**a**) U-Net, (**b**) VGG-16, and (**c**) VGG-19 (The black box shows the difference in predictive ability between the models and white box is the zoomed in view of the black box).

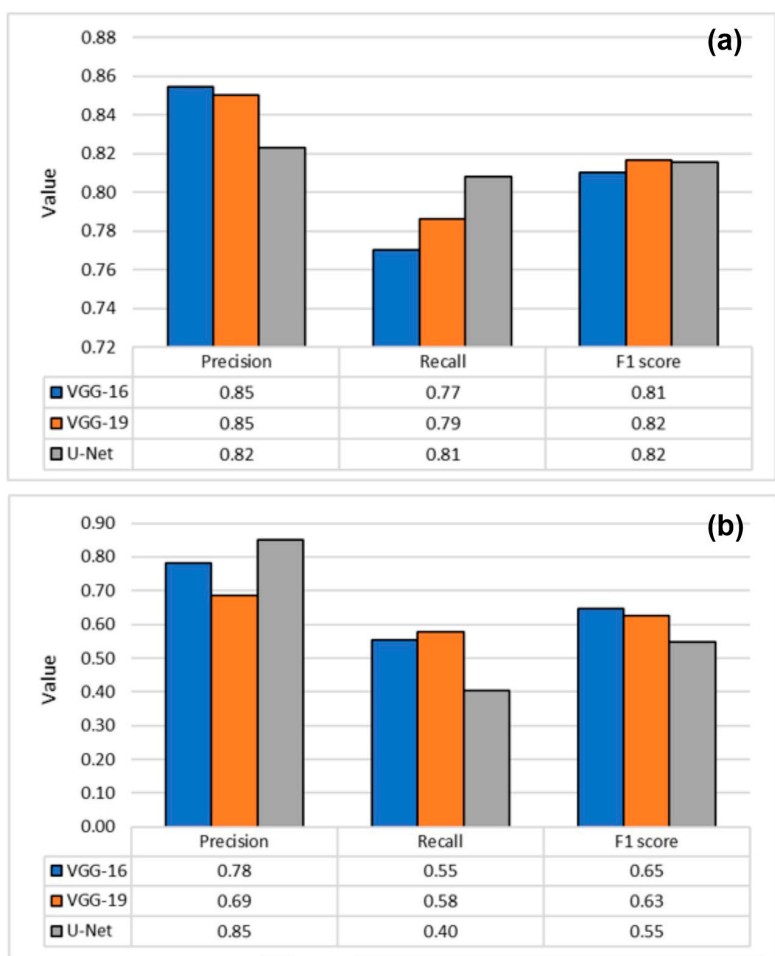

**Figure 10.** Performance evaluation of U-Net, VGG-16, and VGG-19 in the (**a**) training and (**b**) test steps.

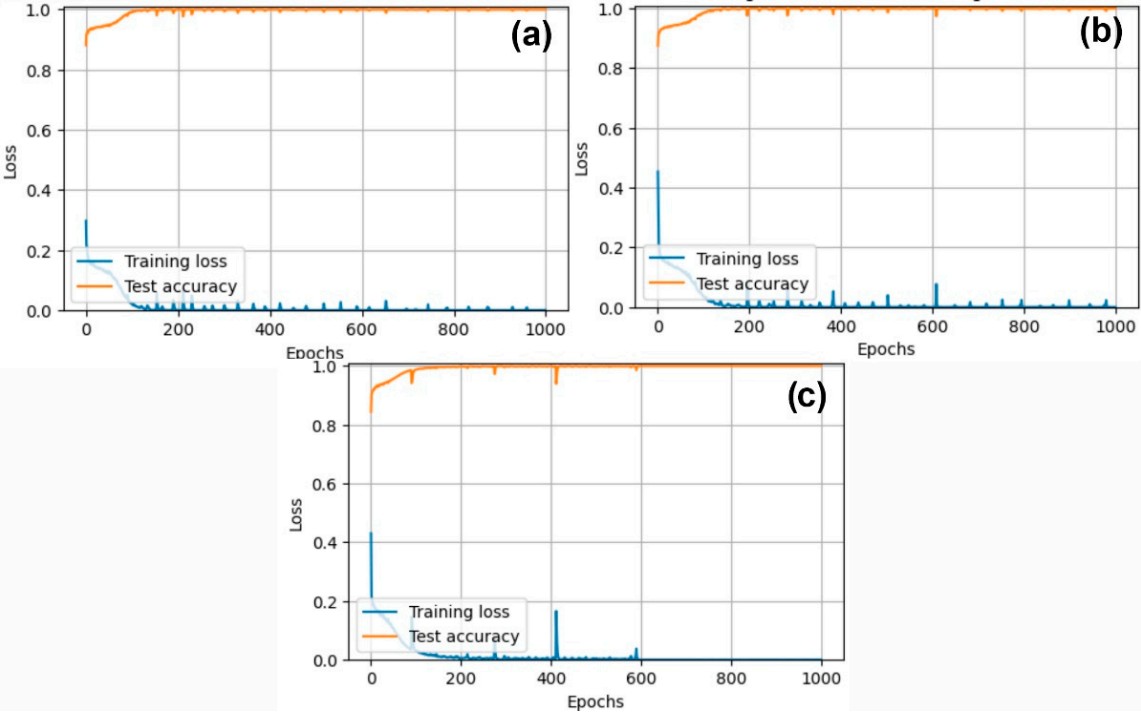

**Figure 11.** Validation loss curves for (**a**) U-Net, (**b**) VGG-16, and (**c**) VGG-19 algorithms.

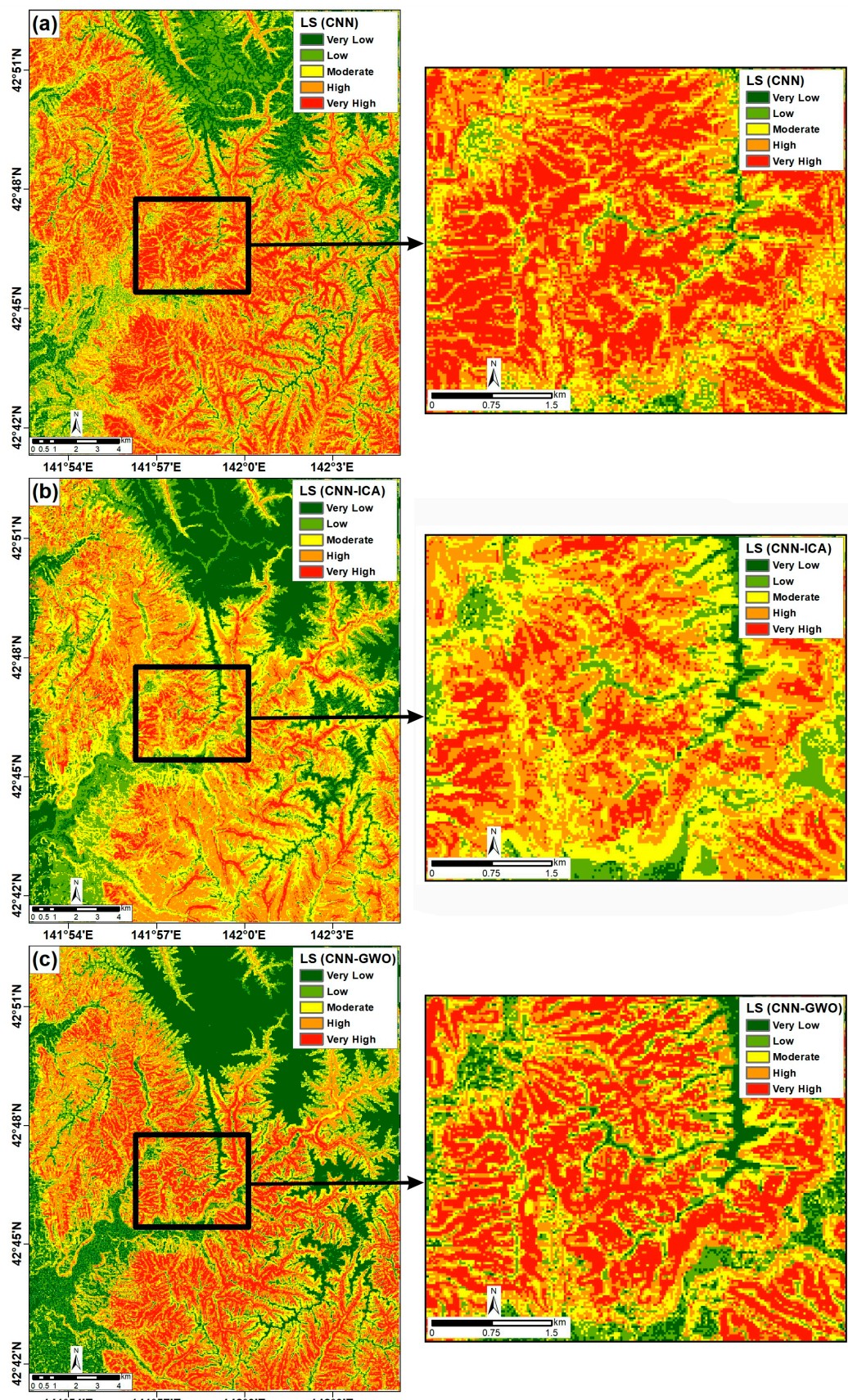

**Figure 12.** Earthquake-induced landslide susceptibility maps created by (**a**) CNN, (**b**) CNN-ICA, and (**c**) CNN-GWO.

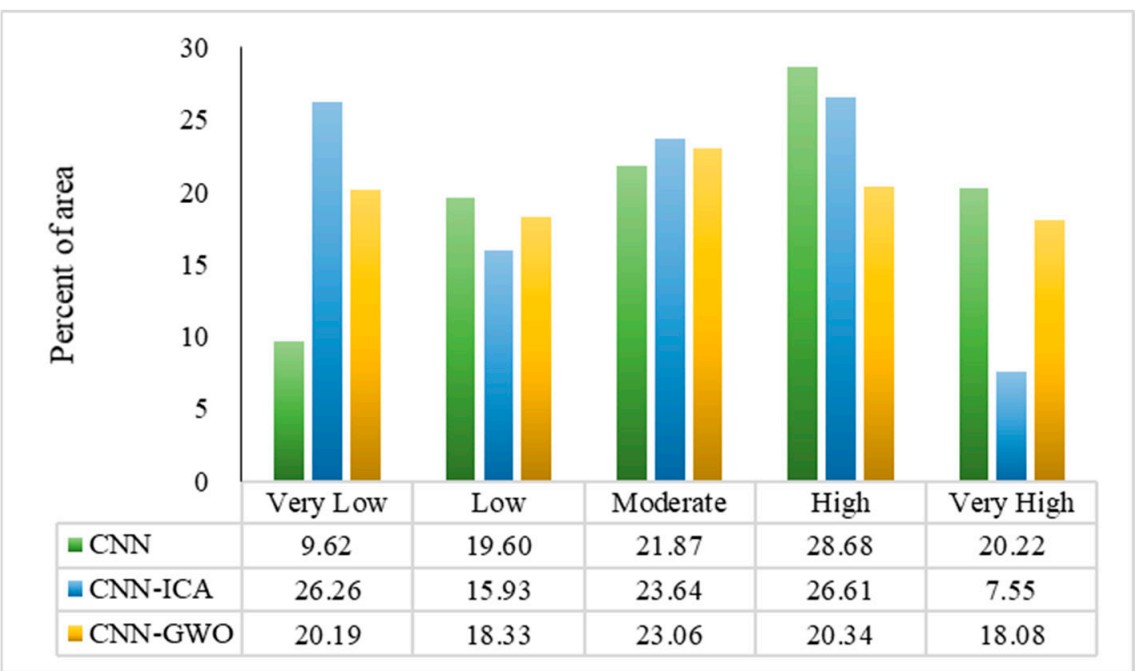

| | Very Low | Low | Moderate | High | Very High |
|---|---|---|---|---|---|
| ■ CNN | 9.62 | 19.60 | 21.87 | 28.68 | 20.22 |
| ■ CNN-ICA | 26.26 | 15.93 | 23.64 | 26.61 | 7.55 |
| ■ CNN-GWO | 20.19 | 18.33 | 23.06 | 20.34 | 18.08 |

**Figure 13.** Proportion of areas of different classes in susceptibility maps derived from CNN, CNN-ICA, and CNN-GWO.

*3.4. Model Performance*

The accuracy of the landslide susceptibility maps is evaluated using RMSE and AUROC. The results from all three methods are evaluated statistically, and their outcomes are compared.

The generalization error of a model is estimated by the mean square error (MSE), and the forecasting errors are measured by the RMSE. An effective model is characterized by a low MSE and RMSE. Figure 14 shows the assessment using both training and test datasets. Two plots are presented for each method: error vs. number of samples and frequency vs. errors. The plot of the error vs. number of samples displays the values of the MSE and RMSE, whereas the plot of the frequency vs. errors illustrates the error mean and standard deviation (StD). For the CNN in the training phase, the MSE, RMSE, error mean, and error StD are 0.097, 0.312, 0.029, and 0.310, respectively. The corresponding values for the CNN-ICA model are 0.094, 0.307, −0.083, and 0.296, and those for the CNN-GWO model are 0.082, 0.286, 0.053, and 0.281. Overall, in the training phase, the CNN-GWO incurs lower errors than the CNN and CNN-ICA. In the case of the test datasets, Among the three methods, the lowest MSE (0.080), RMSE (0.284), and StD (0.280) values correspond to CNN-GWO. As it achieves the lowest errors in both the training and test phases, the CNN-GWO is more accurate and reliable than the other two methods.

Next, the landslide susceptibility maps are assessed using AUROC, as shown in Figure 15. The CNN-GWO method exhibits the highest goodness-of-fit accuracy (0.84), followed by the CNN-ICA model (0.83) and CNN model (0.74) during the training phase. Furthermore, the CNN-GWO method exhibits the highest AUROC of 0.84, followed by the CNN-ICA (0.81) and CNN (0.73). The AUROC results align with the MSE, RMSE, mean error, and standard deviation error values obtained during model validation in both the training and validation phases. Overall, the CNN-GWO model outperforms CNN and CNN-ICA in predicting landslide locations.

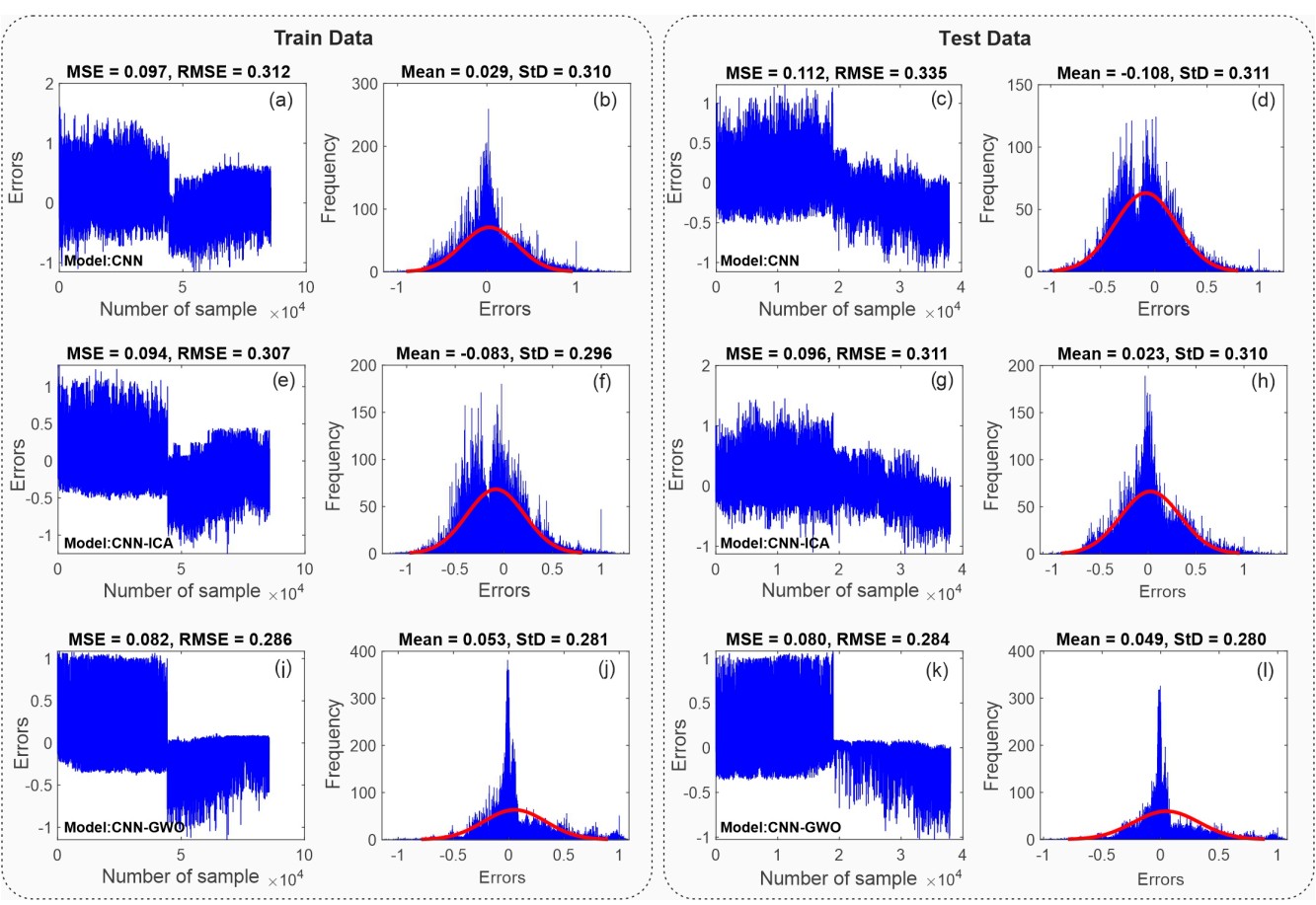

**Figure 14.** Analysis of errors of the landslide susceptibility models (i.e., (**a–d**) CNN, (**e–h**) CNN-ICA, and (**i–l**) CNN-GWO) using the training (**left side**) and testing (**right side**) datasets.

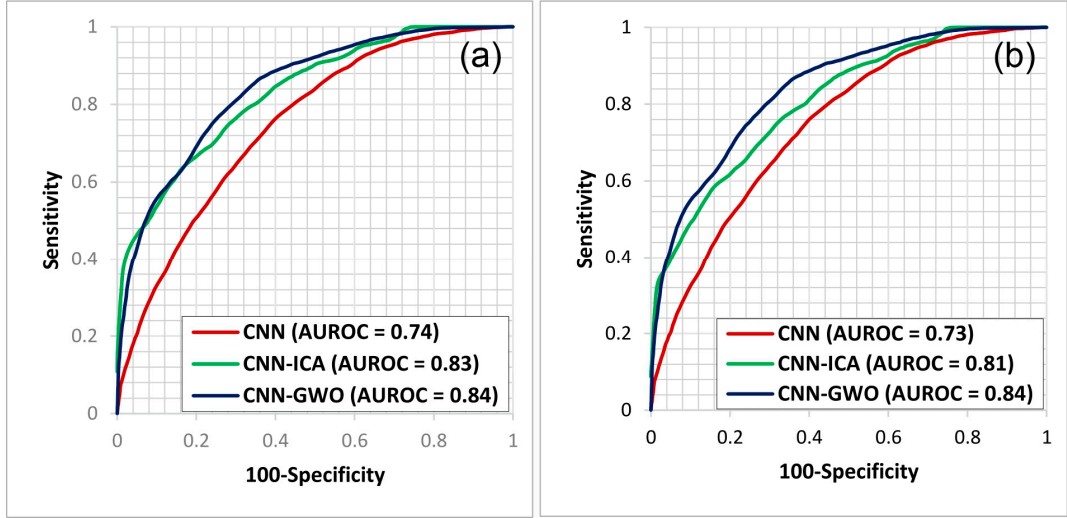

**Figure 15.** Goodness-of-fit and prediction accuracy using (**a**) training and (**b**) testing datasets.

## 4. Discussion

The detection of landslide-prone areas is crucial for disaster management and mitigation. The objective of this study is to identify regions susceptible to landslides by using standard and optimized CNN coupled with the ICA and GWO algorithms. The generation of an inventory map is the first step toward developing a DL model. Landslide inventories

offer essential information for analyzing the deformation mechanisms of landslide incidents and causative factors [98]. The map can be constructed through various approaches, including field surveys, RS, and aerial photography. In field surveys, experts examine the terrain to identify signs of past landslides, such as scars, debris, and slope morphology. However, this task is time-consuming and laborious. Therefore, methods based on the analysis of RS imageries using DL methods have been proposed. Conventional image processing techniques typically depend on manually created features, such as color histograms or texture descriptors, which cannot fully capture the significant information contained in an image [127]. Additionally, these techniques are often sensitive to changes in the input image, such as in lighting or orientation. Moreover, conventional approaches typically require users to process and prepare input images, such as by noise reduction or image segmentation, which may be laborious and error-prone [128]. Because these approaches depend on manual feature engineering, they may not be able to be scaled to high-dimensional image data or complex tasks. In addition, the techniques rely on fixed, predefined feature representations and exhibit limited flexibility given that they are designed to function with certain types of images [129]. In real-world settings, these limitations may deteriorate the accuracy and predictive performance. Because ML algorithms can efficiently analyze and extract characteristics from large datasets of images, identify patterns, and classify images, they are often employed for image processing [130]. DL algorithms, as a subset of the ML algorithm, can learn complicated properties in images, including textures, edges, and shapes, by processing multiple layers of neural networks. Thus, DL algorithms can automatically learn hierarchical representations of image data, resulting in improved accuracy and reliability of results [131]. Furthermore, DL algorithms may be trained end-to-end: The input image is directly fed to the neural network, and the outputs (prediction or classification) are generated without human intervention. This eliminates the need for preprocessing and feature engineering, which are time-consuming and difficult [132]. Furthermore, DL algorithms can be applied in scenarios with limited labeled data. DL algorithms can leverage pre-trained models that have been trained on large-scale datasets. This configuration allows the model to promptly learn new tasks with smaller datasets [133]. Considering these aspects, in this study, three DL algorithms, U-Net, VGG-16, and VGG-19, are used to generate landslide inventory maps from Sentinel-2 data. According to the F1-score, VGG-16 outperforms the other models in predicting landslide locations. The superior performance of VGG-16 compared with U-Net and VGG-19 may be attributable to its deeper architecture and larger number of convolutional layers, which allow it to more effectively extract more complex and high-level features from the input images and better discriminate between landslide and non-landslide areas. Nevertheless, increasing the number of convolutional layers in VGG-19 may lead to the analysis of redundant features in the learning process and overfitting [134]. Li et al. [98] compared the performances of DL models with different structures (e.g., VGG-19, VGG-16, ResNet50, U-Net, DenseNet120, and ResUNet) in detecting seismic secondary landslides in the Wenchuan earthquake area, China. The results showed that the performance of VGG-19 was inferior to that of VGG-16, and it introduced more uncertainties in landslide detection. Moreover, the use of U-Net has been recommended for pixel-based landslide detection when adequate samples are not available. Qin et al. [135] analyzed the predictive ability of distant domain transfer learning, CNN, VGG-16, and VGG-19 for landslide detection in Guangdong Province, China. VGG-16 and VGG-19, as pre-trained models, achieved higher accuracies of 87.09% and 88.24%, respectively, compared with the other two models.

According to the AUROC, MSE, RMSE, and StD values obtained in the present study, the prediction accuracy of the standard CNN is improved by using optimization algorithms. The GWO algorithm is noted to be more effective than ICA in selecting optimum values for the CNN hyperparameters in both the training and testing phases. The individual search behavior and group hunting behavior in GWO enable the exploration of a larger part of the search space, resulting in a better balance between exploration and exploitation compared with that in ICA. Thus, the CNN hyperparameters can be better tuned by the GWO [64].

Additionally, GWO has fewer parameters to tune compared to the ICA, which makes it easier to implement and less prone to overfitting [136]. The results are consistent with those reported by Nosratabadi et al. [137] and Nur et al. [138].

The susceptibility maps indicate that the central regions are the most susceptible to landslides, and the slope failure possibility is high in steep areas with high levels of rainfall along fault lines, rivers, and roads. This finding is consistent with those of previous studies that have highlighted the significant impact of anthropogenic activities, temperature variation, vegetation structures, soil moisture, and topographic nature on landslide occurrence [139–141]. The CNN-ICA is conservative compared to the CNN and CNN-GWO, as it identifies 7.55% and 26.26% of the study area as having very high and very low landslide susceptibility, respectively. In comparison, CNN is more aggressive and classifies 20.22% of the area as having very high susceptibility.

Similar to previous studies, this study involves certain uncertainties attributable to the selection and subjective classification of landslide conditioning factors, volume of datasets, spatial distribution of training and testing samples, resolution/scale of conditioning factors, and hyperparameter selection of DL models. Future studies can be aimed at performing sensitivity and robustness analyses to identify the effect of classification of historical data (70:30 for training and testing) and changes in input parameters on the reliability and accuracy of the susceptibility model. Moreover, other DL algorithms (e.g., ResNet, AlexNet, region-based CNN, and SegNet) can be used to extract historical landslide locations and develop models for susceptibility mapping. Furthermore, other metaheuristic algorithms (e.g., ant colony optimization, cuckoo optimization algorithm, and firefly algorithm) can be used to appropriately tune the hyperparameters of the DL model. Understanding the limitations, strengths, and uncertainties of models can help policymakers make better-informed decisions regarding land-use planning, disaster mitigation, and other critical aspects.

## 5. Conclusions

This research involved two main phases: identification of earthquake-triggered landslide-prone areas and the use of this information to train DL models and produce maps indicating the susceptibility of landslides. To the best of our knowledge, the U-Net, VGG-16, and VGG-19 algorithms have not been widely used for detecting landslides triggered by earthquakes. Therefore, the objective of this study was to assess the effectiveness of these algorithms in evaluating landslide susceptibility. This was achieved through the following steps:

- Landslide inventory locations were extracted from Sentinel satellite imagery related to the Iburi earthquake (2018) in northern Japan using U-Net, VGG-16, and VGG-19.
- Three DL methods, i.e., CNN, CNN-ICA, and CNN-GWO, were used to map the landslide-prone areas.
- The outcomes were evaluated using the precision, recall, and F1-score. VGG-16 yielded the most accurate inventory map. Therefore, it was used in landslide susceptibility mapping.
- The precision and reliability of the landslide susceptibility maps were evaluated using AUROC and RMSE. CNN-GWO, with the lowest MSE (0.080), RMSE (0.284), and error StD (0.280) values and the highest goodness-of-fit and prediction accuracies (0.84), was noted to yield the most reliable outcomes.

Overall, DL methods can be used to effectively assess landslide susceptibility and realize land-use planning and hazard management in areas prone to landslides.

**Author Contributions:** M.S., F.R., H.Ö. and F.S.; writing—review and editing—original draft preparation, S.M.B. and C.J.: project administration, E.H.: review and editing final draft, M.P. and H.M.: Conceptualization, methodology, data curation, formal analysis. All authors have read and agreed to the published version of the manuscript.

**Funding:** This research was supported by the Basic Research Project of the Korea Institute of Geoscience and Mineral Resources (KIGAM) and the National Research Foundation of Korea (NRF) grant funded by the Korean government (MSIT) (No. 2023R1A2C1003095).

**Data Availability Statement:** Raw datasets are publicly available from Google Earth Engine. Our processed data are available upon reasonable request to the corresponding authors.

**Acknowledgments:** We thank the editors and reviewers for providing valuable comments that have helped enhance this manuscript.

**Conflicts of Interest:** The authors declare no conflict of interest.

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
