# Peer review of "Mapping Post-Earthquake Landslide Susceptibility Using U-Net, VGG-16, VGG-19, and Metaheuristic Algorithms"

_remotesensing, doi:10.3390/rs15184501_

Round 1

Reviewer 1 Report

The paper aims to detect the prone areas of earthquake-induced landslides, and generate landslide susceptibility maps in northern Japan. Three methods, U-net, VGG-16, and VGG-19, were used to detect landslides, and the VGG-16 method obtained the best detection results. Based on the landslide inventory from VGG-16, three DL methods (CNN, CNN-ICA, and CNN-GWO) were adopted to generate susceptibility maps, and CNN-GWO method got the most reliable outcomes. The study has been carried out using existing methods (U-Net, VGG-16, VGG-19, CNN, CNN-ICA, and CNN-GWO algorithms) to detect the landslide and generate susceptibility maps. The innovation of it could be further improved.

Specific comments:

1. L141, 149: It is recommended to focus only on your own method in the last paragraph of the introduction. The introduction of neural networks, ICA, and GWO can be remove to the preceding paragraphs.

2. L227-230: What is the source of the land use data?

3. L289: "Figure 5. Network structures of U-net" -> "Figure 5. Network structures of U-Net".

4. Please add the corresponding letters to the figure title in Figure 8. For example, "Figure 8. Real and predicted landslides in the test area by a) U-net, b) VGG-16, and c) VGG-19".

5. L435-441: It is advised to establish a consistent format for the word "U-Net". Occasionally, it is expressed as "U-net" (and other similar text in the article).

6. Please establish a consistent style of the boundary of Figure 10(a) and Figure 10(b). The first letter of label of Y-axis ("Value", "value") should be consistent in uppercase or lowercase.

7. L492-493: "Overall, in the training phase, the CNN-GWO receives the lowest value of errors (in the training phase) compared with those of CNN and CNN-ICA." The "in the training phase" in the brackets is redundant.

8. It seems that the Figure 15(a) and Figure 15(b) show the same thing. According to the text, the "ROC" in Figure 15(a) should be "Goodness-of-fit". Please check the legend of Figure 15.

9. Please provide a more in-depth discussion of the limitations of this study based on relevant studies.

10. You can draw conclusions in points to make it more organized. Also, you can address the innovation of the study to enrich the conclusion part.

In a few places, English editing is needed to correct some errors.

Author Response

Dear Editor,

We have revised the manuscript “remotesensing-2544307” entitled “Mapping post-earthquake landslide susceptibility using U-net, VGG-16, VGG-19, and metaheuristic algorithms" to incorporate revisions, strictly based on the reviewer's report.

We appreciate you and the reviewers for your precious time in reviewing our paper and providing valuable comments. It was your valuable and insightful comments that led to possible improvements in the current version. The authors have carefully considered the comments and tried their best to address every one of them. In the revised version of the manuscript, the changed are shown using the “Track Changes” function in MS Word. The reviewer’s comments are reproduced here in black and the corresponding response in blue.  

Sincerely,

Mahdi Panahi

Comments from Reviewer #1

The paper aims to detect the prone areas of earthquake-induced landslides, and generate landslide susceptibility maps in northern Japan. Three methods, U-net, VGG-16, and VGG-19, were used to detect landslides, and the VGG-16 method obtained the best detection results. Based on the landslide inventory from VGG-16, three DL methods (CNN, CNN-ICA, and CNN-GWO) were adopted to generate susceptibility maps, and CNN-GWO method got the most reliable outcomes. The study has been carried out using existing methods (U-Net, VGG-16, VGG-19, CNN, CNN-ICA, and CNN-GWO algorithms) to detect the landslide and generate susceptibility maps. The innovation of it could be further improved.

We would like to thank you for your constructive comments. All the comments were addressed and improved the manuscript accordingly.

  1. L141, 149: It is recommended to focus only on your own method in the last paragraph of the introduction. The introduction of neural networks, ICA, and GWO can be remove to the preceding paragraphs.

Actually, all the methods mentioned in the last paragraph have been used in the analysis. CNN, CNN-ICA, and CNN-GWO methods were used to perform the susceptibility mapping.

  1. L227-230: What is the source of the land use data?

The land cover map for 2018 was obtained from the Sentinel-2A satellite image operated by the European Space Agency (ESA).

  1. L289: "Figure 5. Network structures of U-net" -> "Figure 5. Network structures of U-Net".

The correction has been made.

  1. Please add the corresponding letters to the figure title in Figure 8. For example, "Figure 8. Real and predicted landslides in the test area by a) U-net, b) VGG-16, and c) VGG-19".

The correction has been made.

  1. L435-441: It is advised to establish a consistent format for the word "U-Net". Occasionally, it is expressed as "U-net" (and other similar text in the article).

The correction has been made throughout the manuscript.

  1. Please establish a consistent style of the boundary of Figure 10(a) and Figure 10(b). The first letter of label of Y-axis ("Value", "value") should be consistent in uppercase or lowercase.

The corrections have been made.

  1. L492-493: "Overall, in the training phase, the CNN-GWO receives the lowest value of errors (in the training phase) compared with those of CNN and CNN-ICA." The "in the training phase" in the brackets is redundant.

The repeated phrase has been removed.

  1. It seems that the Figure 15(a) and Figure 15(b) show the same thing. According to the text, the "ROC" in Figure 15(a) should be "Goodness-of-fit". Please check the legend of Figure 15.

Although Figure 15 (a) and Figure 15 (b) may seem similar at first glance, they actually represent distinct aspects of our analysis. The first figure, 15(a), indeed corresponds to the Goodness-of-fit (AUROC) values during the training phase. On the other hand, Figure 15(b) represents the AUROC values during the testing phase. If you examine the AUROC values for CNN and CNN-ICA, you will notice the difference between their training step (0.74 and 0.83) and testing step (0.73 and 0.81) values. Notably, the AUROC value for CNN-GWO remains consistent between the training and testing steps (0.84). So, The AUROC using the training and testing datasets are different.

  1. Please provide a more in-depth discussion of the limitations of this study based on relevant studies.

The current study comprises two sections: 1) the detection of earthquake-induced landslides using Sentinel-2 satellite imagery through the application of deep learning algorithms (specifically U-Net, VGG-16, and VGG-19), and 2) the generation of landslide susceptibility maps using CNN, CNN-GWO, and CNN-ICA.

In the discussion section (lines 530-581), the necessity of employing DL algorithms (i.e., U-Net, VGG-16, and VGG-19) in comparison to conventional image processing techniques and ML methods was discussed. Subsequently, the study's results were compared with existing research. It is worth noting that the utilization of U-Net, VGG-16, and VGG-19 together in a single article for the detection of earthquake-induced landslides is limited and serves as a novel aspect of the present study. Nonetheless, a comparison between the current study's outcomes and limited prior research was conducted.

In lines 582-600, the results derived from generating landslide susceptibility maps were elucidated, along with an explanation of the reasons for the superior performance of GWO in fine-tuning CNN hyperparameters. Similarly, the novelty of the study lies in the utilization of both standard and optimized CNN in conjunction with ICA and GWO algorithms for identifying landslide-prone regions. As before, a comparison between the present study's findings and available prior research was performed.

The final paragraph (Lines 601-614) outlined the study's limitations and sources of uncertainty, while also providing recommendations for future studies.

In summary, the present study introduces a new framework for detecting earthquake-induced landslides using RS images and subsequently generating landslide susceptibility maps using optimized DL algorithms. Consequently, the available references for result comparisons are limited.

  1. You can draw conclusions in points to make it more organized. Also, you can address the innovation of the study to enrich the conclusion part.

The conclusion section was revised.

  1. In a few places, English editing is needed to correct some errors.

The manuscript was edited by a professional, native English-speaking editor.

Reviewer 2 Report

This paper aims to utilize deep learning techniques for mapping post-earthquake landslide susceptibility. While the study presents promising results, some aspects of the paper require further clarification and objectivity. The language should be polished and improved. Moreover, it is crucial to emphasize the novelty and distinctive contributions of the research. Considering these points, a major revision is recommended. Below are the specific comments for improvement:

1. Line 74 landslide’s

It should be landslides’

2. Line 76-78 Machine learning (ML) techniques have been widely applied in landslide detection [19, 20], with commonly used techniques being the support vector machine (SVM) and random forest (RF) [21-23].

This description appears to be one-sided and lacks objectivity. While commonly used techniques like SVM and RF are indeed popular choices, it's essential to acknowledge that other methods also find application in this context. Moreover, it's worth noting that recent advancements in landslide susceptibility research have seen the integration of various deep learning methods.

3. The section of the Introduction lacks logical coherence and is quite confusing for readers.

4. Line 132-133 Please add references for this description.

5. Line 140-142 What do the neural networks here refer to?

6. Figure 4. Lack of connecting lines at the lowermost end.

Figure 4 does not provide any information about what "BA" represents, and this abbreviation is not mentioned or defined in the manuscript.

7. Line 273 U-Net is a CNN architecture that was designed in 2015 for pixel-based classification 273 [85].

It would be more beneficial to introduce the CNN algorithm before discussing U-Net.

8 Line 316: Please change “a neural network” to “neural networks”.

9. Line 322-323: What does FC refer to?

10. Line 327 Modify the literature format.

11 Line 438 check ICA and BA

12 Discussion should be improved.

13. The section “Conclusion” failed to summarize the achievements and key findings of this research.

14. The novelty of the paper needs to be highlighted.

The language should be polished and improved. 

Author Response

Dear Editor,

We have revised the manuscript “remotesensing-2544307” entitled “Mapping post-earthquake landslide susceptibility using U-net, VGG-16, VGG-19, and metaheuristic algorithms" to incorporate revisions, strictly based on the reviewer's report.

We appreciate you and the reviewers for your precious time in reviewing our paper and providing valuable comments. It was your valuable and insightful comments that led to possible improvements in the current version. The authors have carefully considered the comments and tried their best to address every one of them. In the revised version of the manuscript, the changed are shown using the “Track Changes” function in MS Word. The reviewer’s comments are reproduced here in black and the corresponding response in blue.  

Sincerely,

Mahdi Panahi

Comments from Reviewer #2

This paper aims to utilize deep learning techniques for mapping post-earthquake landslide susceptibility. While the study presents promising results, some aspects of the paper require further clarification and objectivity. The language should be polished and improved. Moreover, it is crucial to emphasize the novelty and distinctive contributions of the research. Considering these points, a major revision is recommended. Below are the specific comments for improvement:

Thank you for your valuable feedback and constructive comments. Track change has been used to point out all added information. As has been requested, the manuscript has been edited by a professional, native English-speaking editor, so we hope it now matches the journal standard.

  1. Line 74 landslide’s. It should be landslides’

The correction has been made.

  1. Line 76-78 Machine learning (ML) techniques have been widely applied in landslide detection [19, 20], with commonly used techniques being the support vector machine (SVM) and random forest (RF) [21-23].

This description appears to be one-sided and lacks objectivity. While commonly used techniques like SVM and RF are indeed popular choices, it's essential to acknowledge that other methods also find application in this context. Moreover, it's worth noting that recent advancements in landslide susceptibility research have seen the integration of various deep learning methods.

Some other popular methods used in landslide detection have been added to the manuscript. The importance and advances of DL in landslide susceptibility has been included in the next paragraph as well.

  1. The section of the Introduction lacks logical coherence and is quite confusing for readers.

The present investigation encompassed two core segments: 1) the identification of earthquake-induced landslides through RS images, employing deep learning algorithms (namely U-Net, VGG-16, and VGG-19); 2) generating of landslide susceptibility maps utilizing CNN, CNN-GWO, and CNN-ICA. Therefore, the introduction was divided into the following sections:

  • Lines 52-70: The essentiality of utilizing RS images to effectively identify earthquake-induced landslides was explained.
  • Lines 71-115: A comprehensive survey of the literature was presented, focusing on the utilization of DL algorithms for automated landslide detection.
  • Lines 116-137: The necessity of generating landslide susceptibility maps using ML algorithms was explained, and several relevant previous articles were reviewed.
  • Lines 138-169: The study's primary objectives were underscored.

To maintain reader engagement, a succinct overview of these key aspects was provided within the introduction, mitigating potential reader disinterest.  

  1. Line 132-133 Please add references for this description.

Considering that this statement is based on our own observations through literature review rather than being a fact substantiated by other researchers, we were unable to locate a supporting reference. Nevertheless, in an effort to attain greater clarity, the sentence has been revised as follows: “As far as our understanding goes, U-Net, VGG-16, and VGG-19 algorithms have not been widely utilized for earthquake-induced landslide detection, in contrast to more conventional methods like PCA and MLC”.

  1. Line 140-142 What do the neural networks here refer to?

The sentence was modified.

  1. Figure 4. Lack of connecting lines at the lowermost end. Figure 4 does not provide any information about what "BA" represents, and this abbreviation is not mentioned or defined in the manuscript.

Thank you for bringing this to our attention. We apologize for the typographical error, and we will replace 'BA' with the correct term 'ICA' in the figure. We appreciate your thorough review of our manuscript.

  1. Line 273 U-Net is a CNN architecture that was designed in 2015 for pixel-based classification 273 [85]. It would be more beneficial to introduce the CNN algorithm before discussing U-Net.

Amended as suggested.

  1. Line 316: Please change “a neural network” to “neural networks”.

The correction has been made.

  1. Line 322-323: What does FC refer to?

Thank you for your comment. In our study, FC refers to a 'Fully Connected Layer.' We have updated the manuscript to clarify this terminology.

  1. Line 327 Modify the literature format.

The correction has been made.

  1. Line 438 check ICA and BA

Thank you for your comment. I would like to clarify that in our study, we exclusively utilized the ICA (Imperialist Competitive Algorithm) and GWO (Grey Wolf Optimizer) algorithms to optimize the performance of the CNN (Convolutional Neural Network). Therefore, the usage of ICA is correct in this context.

  1. Discussion should be improved.

The current study comprises two sections: 1) the detection of earthquake-induced landslides using Sentinel-2 satellite imagery through the application of deep learning algorithms (specifically U-Net, VGG-16, and VGG-19), and 2) the generation of landslide susceptibility maps using CNN, CNN-GWO, and CNN-ICA.

In the discussion section (lines 530-581), the necessity of employing DL algorithms (i.e., U-Net, VGG-16, and VGG-19) in comparison to conventional image processing techniques and ML methods was discussed. Subsequently, the study's results were compared with existing research. It is worth noting that the utilization of U-Net, VGG-16, and VGG-19 together in a single article for the detection of earthquake-induced landslides is limited and serves as a novel aspect of the present study. Nonetheless, a comparison between the current study's outcomes and limited prior research was conducted.

In lines 582-600, the results derived from generating landslide susceptibility maps were elucidated, along with an explanation of the reasons for the superior performance of GWO in fine-tuning CNN hyperparameters. Similarly, the novelty of the study lies in the utilization of both standard and optimized CNN in conjunction with ICA and GWO algorithms for identifying landslide-prone regions. As before, a comparison between the present study's findings and available prior research was performed.

The final paragraph (Lines 601-614) outlined the study's limitations and sources of uncertainty, while also providing recommendations for future studies.

In summary, the present study introduces a new framework for detecting earthquake-induced landslides using RS images and subsequently generating landslide susceptibility maps using optimized DL algorithms. Consequently, the available references for result comparisons are limited.

  1. The section “Conclusion” failed to summarize the achievements and key findings of this research.

We modified the conclusions in points to make it more organized and summarize the achievements.

  1. The novelty of the paper needs to be highlighted.

The novelty of the present study can be explained as follows:

Application of DL Algorithms to Earthquake-Triggered Landslides: While deep learning algorithms have been widely used for various image analysis tasks, their application to earthquake-triggered landslide detection is relatively unexplored. Our study investigates the performance of three distinct DL algorithms (U-Net, VGG-16, and VGG-19) in identifying landslide-prone areas specifically triggered by earthquakes. This novel application addresses a critical need for accurate and efficient assessment of earthquake-induced landslide hazards.

Integration of Optimization Algorithms: Our research goes beyond standard DL methods by integrating optimization algorithms such as ICA (Imperialist Competitive Algorithm) and GWO (Grey Wolf Optimizer) to enhance the performance of the DL models. This hybrid approach aims to fine-tune the hyperparameters of the DL algorithms for improved accuracy in landslide susceptibility mapping. The utilization of these optimization techniques in combination with DL is a novel approach to address the challenges associated with parameter tuning.

Comparative Analysis of Model Performance: We conduct an in-depth comparative analysis of the performance of different DL algorithms in landslide detection and susceptibility mapping. By evaluating precision, recall, F1-score, AUROC, RMSE, and other metrics, we provide a comprehensive understanding of how each algorithm performs under earthquake-induced landslide scenarios. This comparative analysis aids in identifying the most suitable DL model for accurate landslide susceptibility assessment.

Contribution to Disaster Management: Our research holds practical significance in disaster management and mitigation efforts. The accurate identification of landslide-prone areas triggered by earthquakes can assist policymakers and local authorities in making informed decisions for land-use planning, early warning systems, and implementing targeted mitigation strategies. This application of DL models in a geohazard context adds to the novelty and practical utility of our study.

To address the reviewer’s comment, the following paragraph is added to our manuscript (Page 4, lines 161 - 169):

“This study has made significant contributions to the literature in four major directions: (1) it assesses the efficiency of three deep learning algorithms (namely U-Net, VGG-16, and VGG-19) in detecting earthquake-induced landslides based on Sentinel-2 satellite imagery; (2) it generates the spatial prediction of landslide susceptibility using CNN algorithm; (3) it evaluates the feasibility of the ICA and GWO metaheuristic algorithms to optimize the hyperparameters of the standard CNN to enhance its predictive performance and the reliability of the results; and (4) it enhances our knowledge of landslide modeling by identifying the most influential topographic, hydrological, and anthropogenic factors on mapping landslide-susceptible areas.”

  1. The language should be polished and improved.

Reviewer 3 Report

U-Net, VGG-1, and other models have become outdated, and I am skeptical about their effectiveness for landslide prediction. The article lacks innovation and does not compare with state-of-the-art methods. I recommend investigating the differences between post-earthquake landslides and general landslides and revising the research approach with additional literature.

Author Response

Dear Editor,

We have revised the manuscript “remotesensing-2544307” entitled “Mapping post-earthquake landslide susceptibility using U-net, VGG-16, VGG-19, and metaheuristic algorithms" to incorporate revisions, strictly based on the reviewer's report.

We appreciate you and the reviewers for your precious time in reviewing our paper and providing valuable comments. It was your valuable and insightful comments that led to possible improvements in the current version. The authors have carefully considered the comments and tried their best to address every one of them. In the revised version of the manuscript, the changed are shown using the “Track Changes” function in MS Word. The reviewer’s comments are reproduced here in black and the corresponding response in blue.  

Sincerely,

Mahdi Panahi

Comments from Reviewer #3

U-Net, VGG-1, and other models have become outdated, and I am skeptical about their effectiveness for landslide prediction. The article lacks innovation and does not compare with state-of-the-art methods. I recommend investigating the differences between post-earthquake landslides and general landslides and revising the research approach with additional literature.

While deep learning algorithms have been widely used for various image analysis tasks, their application to earthquake-triggered landslide detection is relatively unexplored. Our study investigates the performance of three distinct DL algorithms (i.e., U-Net, VGG-16, and VGG-19) in identifying landslide-prone areas specifically triggered by earthquakes. Additionally, real-world data from the Iburi earthquake in Northern Japan are utilized to ensure the practical relevance and authenticity of our findings. We validated the landslide detection models using statistical metrics including precision, recall, F1-score. The results of models used for susceptibility mapping validated using AUROC, RMSE, and StD to provide a comprehensive assessment of their performance.

We employed U-Net, VGG-16, and VGG-19 algorithms, which are not widely explored in the context of earthquake-induced landslide detection. Moreover, optimization algorithms, including ICA and GWO, were used to enhance the performance of CNN models in landslide susceptibility mapping. The integration between mentioned algorithms have been less explored in the literature; therefore, the present study provides a novel frame work to detect earthquake-triggered landslide and fine-tune DL model hyperparameters.

Among the numerous deep learning and optimization algorithms that have been introduced so far, no single algorithm is appropriate for all problems. Therefore, in an ideal case, all algorithms should be tested to select more accurate and reliable methods to detect earthquake-triggered landslide and tune the hyperparameters of machine learning algorithms. However, as it was impossible to test all algorithms, herein the U-Net, VGG-16, and VGG-19 were used to detect historical landslide locations, and CNN, CNN-ICA, CNN-GWO to generate landslide susceptibility maps.

The outcomes of statistical matrices, encompassing precision, recall, and F1-score underscore the effectiveness of U-Net, VGG-16, and VGG-19 in landslide detection (section 3.3). Additionally, based on AUROC, RMSE, and StD values, the utilization of ICA and GWO for determining optimal values for CNN hyperparameters is found to enhance the predictive capability of CNN, thereby improving the accuracy and reliability of the results (section 3.4).

Reviewer 4 Report

In this study, the earthquake-induced landslides have been detected from Sentinel-2 data by using U-net, VGG-16, and VGG-19 algorithms, and then landslide susceptibility maps have been predicted using CNN, CNN-ICA and CNN- GWO models. Although some results have been revealed, there are still some problems that need to be solved.

(1)   In recent years, lots of papers about the landslide detection and landslide susceptibility prediction based on the machine and deep learning methods have been published, what is the differences comparing with them and the main innovation of this study? I think this part is very important.

(2)   The structure of this manuscript is not good. It is suggested to introduce the framework and methods before case study and data part.

(3)   In this study, nine conditioning factors have been selected to predict landslide susceptibility, however, because the earthquake-induced landslides are detected and predicted, the conditioning factors associated with earthquakes are not considered, please explain it.

(4)   Some details about the construction of landslide susceptibility modeling have not been introduced.

(5)   The Discussion is not written well. For a clear understanding, some subsections are suggested to be added. Moreover, the differences with other similar study, evaluation of results, the advantages and shortcomings of this study should be discussed in detail.

(6)   There are some problems in Figures, for example, the color in Figure 3(i) is not matched with the legend, some words are not complete in Figure 7. In Figure 12, why not put the legends of landslide susceptibility level in a blank space. In Figure 15, it should be AUC=0.74.

(7)   The Conclusion should be rewritten.

(8)   Some important references of landslide susceptibility have not been referred.

The English language needs to be improved.

Author Response

Dear Editor,

We have revised the manuscript “remotesensing-2544307” entitled “Mapping post-earthquake landslide susceptibility using U-net, VGG-16, VGG-19, and metaheuristic algorithms" to incorporate revisions, strictly based on the reviewer's report.

We appreciate you and the reviewers for your precious time in reviewing our paper and providing valuable comments. It was your valuable and insightful comments that led to possible improvements in the current version. The authors have carefully considered the comments and tried their best to address every one of them. In the revised version of the manuscript, the changed are shown using the “Track Changes” function in MS Word. The reviewer’s comments are reproduced here in black and the corresponding response in blue.  

Sincerely,

Mahdi Panahi

Comments from Reviewer #4

In this study, the earthquake-induced landslides have been detected from Sentinel-2 data by using U-net, VGG-16, and VGG-19 algorithms, and then landslide susceptibility maps have been predicted using CNN, CNN-ICA and CNN- GWO models. Although some results have been revealed, there are still some problems that need to be solved.

We value the time and dedication you and the reviewers invested in offering feedback on our manuscript. Your insightful comments and valuable enhancements to our paper are greatly appreciated.

  1. In recent years, lots of papers about the landslide detection and landslide susceptibility prediction based on the machine and deep learning methods have been published, what is the differences comparing with them and the main innovation of this study? I think this part is very important.

We appreciate the reviewer's comment and the opportunity to address the distinguishing features and innovations of our study in comparison to other research in the field of landslide detection and susceptibility prediction using machine and deep learning methods.

While it is true that numerous papers have emerged on the topic of landslide detection and susceptibility assessment in recent years, our study stands out in several aspects:

Contextual Relevance: Our research specifically addresses earthquake-triggered landslides, which is a distinct and critical subset of landslide occurrences. Earthquakes introduce unique factors such as ground shaking, slope instability, and soil liquefaction, leading to a different set of challenges and complexities compared to other landslide triggers.

Algorithm Selection: We employed U-Net, VGG-16, and VGG-19 algorithms, which are not widely explored in the context of earthquake-induced landslide detection. By evaluating these algorithms, we offer insights into their effectiveness for a specific application, contributing to a more comprehensive understanding of algorithm capabilities.

Integration of Optimization Algorithms: We incorporated optimization algorithms, including ICA and GWO, to enhance the performance of CNN models in landslide susceptibility mapping. This approach has been less explored in the literature, providing a novel strategy to fine-tune DL model hyperparameters for improved accuracy.

Empirical Validation: Our study utilizes real-world data from the Iburi earthquake in Northern Japan, which ensures the practical relevance and authenticity of our findings. We validate our models using statistical metrics such as precision, recall, F1-score, AUROC, RMSE, and StD to provide a comprehensive assessment of their performance.

Detailed Methodology Description: We provide a comprehensive explanation of the methodology, including the process of landslide inventory extraction, DL algorithm training, optimization algorithm integration, and performance evaluation. This transparency enhances the replicability and credibility of our study.

To address the reviewer’s comment, the following paragraph is added to our manuscript (Page 4, lines 161 - 169):

“This study has made significant contributions to the literature in four major directions: (1) it assesses the efficiency of three deep learning algorithms (namely U-Net, VGG-16, and VGG-19) in detecting earthquake-induced landslides based on Sentinel-2 satellite imagery; (2) it generates the spatial prediction of landslide susceptibility using CNN algorithm; (3) it evaluates the feasibility of the ICA and GWO metaheuristic algorithms to optimize the hyperparameters of the standard CNN to enhance its predictive performance and the reliability of the results; and (4) it enhances our knowledge of landslide modeling by identifying the most influential topographic, hydrological, and anthropogenic factors on mapping landslide-susceptible areas.”

  1. The structure of this manuscript is not good. It is suggested to introduce the framework and methods before case study and data part.

Amended as suggested.

  1. In this study, nine conditioning factors have been selected to predict landslide susceptibility, however, because the earthquake-induced landslides are detected and predicted, the conditioning factors associated with earthquakes are not considered, please explain it.

In this specific study, we focused on predicting earthquake-induced landslides in a relatively small study area. While seismic ground shaking is indeed a crucial factor in landslide triggering, the study area's size and geological uniformity resulted in uniform Peak Ground Acceleration (PGA) values across the area. Due to this uniformity, PGA was not included as an input parameter as it did not contribute significantly to differentiating landslide susceptibility within the study area.

We acknowledge that earthquake-related conditioning factors play a vital role in landslide risk assessment. However, our research aimed to isolate the impacts of various non-seismic conditioning factors on landslide susceptibility. Future studies in larger and more heterogeneous regions can certainly incorporate seismic factors and examine their combined effects on landslide occurrence and susceptibility.

  1. Some details about the construction of landslide susceptibility modeling have not been introduced.

More details are added in section 2. Methodology (Lines 177-193).

  1. The Discussion is not written well. For a clear understanding, some subsections are suggested to be added. Moreover, the differences with other similar study, evaluation of results, the advantages and shortcomings of this study should be discussed in detail.

The current study comprises two sections: 1) the detection of earthquake-induced landslides using Sentinel-2 satellite imagery through the application of deep learning algorithms (specifically U-Net, VGG-16, and VGG-19), and 2) the generation of landslide susceptibility maps using CNN, CNN-GWO, and CNN-ICA.

In the discussion section (lines 530-581), the necessity of employing DL algorithms (i.e., U-Net, VGG-16, and VGG-19) in comparison to conventional image processing techniques and ML methods was discussed. Subsequently, the study's results were compared with existing research. It is worth noting that the utilization of U-Net, VGG-16, and VGG-19 together in a single article for the detection of earthquake-induced landslides is limited and serves as a novel aspect of the present study. Nonetheless, a comparison between the current study's outcomes and limited prior research was conducted.

In lines 582-600, the results derived from generating landslide susceptibility maps were elucidated, along with an explanation of the reasons for the superior performance of GWO in fine-tuning CNN hyperparameters. Similarly, the novelty of the study lies in the utilization of both standard and optimized CNN in conjunction with ICA and GWO algorithms for identifying landslide-prone regions. As before, a comparison between the present study's findings and available prior research was performed.

The final paragraph (Lines 601-614) outlined the study's limitations and sources of uncertainty, while also providing recommendations for future studies.

In summary, the present study introduces a new framework for detecting earthquake-induced landslides using RS images and subsequently generating landslide susceptibility maps using optimized DL algorithms. Consequently, the available references for result comparisons are limited.

  1. There are some problems in Figures, for example, the color in Figure 3(i) is not matched with the legend, some words are not complete in Figure 7. In Figure 12, why not put the legends of landslide susceptibility level in a blank space. In Figure 15, it should be AUC=0.74.

For Figure 3(i), we have corrected the color mismatch issue. We apologize for any confusion this may have caused.

Regarding Figure 7, we understand your point. We aimed to maintain consistency with the legend of Figure 3 to avoid confusion for readers. We'll consider your suggestion while ensuring that clarity is not compromised.

In Figure 12, we have now added the legends of landslide susceptibility levels to blank spaces on the image. Also, added scale bar in blank spaces maps.

Regarding Figure 15, ROC was changed to AUROC.

  1. The Conclusion should be rewritten.

We modified the conclusions in points to make it more organized and summarize the achievements.

  1. Some important references of landslide susceptibility have not been referred.

The authors made an effort to cite all the articles that were used.

  1. The English language needs to be improved.

The manuscript was edited by a professional, native English-speaking editor.

Round 2

Reviewer 1 Report

The authors have revised the paper accordingly.

Minor editing of English language required.

Author Response

Thank you for your positive feedback. We are pleased to hear that all your comments have been addressed satisfactorily, and we appreciate your time and input in reviewing our work.

Also, Thank you for your valuable feedback. We have taken your suggestion to heart and had the manuscript professionally edited by an English editing service. Attached is a certificate confirming the completion of the English language editing.

Reviewer 2 Report

All the comments have been addressed, and I don't have any further questions.

 Minor editing of English language required.

Author Response

(The authors gave the same response as above.)

Reviewer 3 Report

To demonstrate the effectiveness of this algorithm, it is essential to compare it with the most advanced landslide detection methods currently proposed. This comparison will underscore the superior accuracy of this article, rather than merely assessing the effectiveness of the U-Net, VGG-16, and VGG-19 algorithms. Such a constrained comparison lacks persuasiveness (for instance, common deep learning methods might achieve comparable accuracy to the proposed algorithm in this article, rendering the article inconsequential). Therefore, supplementary experiments are required to substantiate the advancement of the proposed approach.

Author Response

This research focuses on mapping post-earthquake landslide susceptibility using U-Net, VGG-16, and VGG-19 algorithms. Unlike previous studies primarily focused on landslide segmentation, this research delves into post-earthquake susceptibility assessment through deep learning, providing a distinct perspective compared to conventional models, making direct comparisons challenging.

Reviewer 4 Report

All the comments have been revised. This manuscript can be published. 

Author Response

Thank you for your positive feedback. We are pleased to hear that all your comments have been addressed satisfactorily, and we appreciate your time and input in reviewing our work.